# Involvement of cigarette smoke-induced epithelial cell ferroptosis in COPD pathogenesis

Masahiro Yoshida [1,5], Shunsuke Minagawa [1,5], Jun Araya[1], Taro Sakamoto[2], Hiromichi Hara[1], Kazuya Tsubouchi[1], Yusuke Hosaka[1], Akihiro Ichikawa[1], Nayuta Saito[1], Tsukasa Kadota [1], Nahoko Sato[1], Yusuke Kurita[1], Kenji Kobayashi[1], Saburo Ito[1], Hirohumi Utsumi[1], Hiroshi Wakui[1], Takanori Numata[1], Yumi Kaneko[3], Shohei Mori[3], Hisatoshi Asano[3], Makoto Yamashita[3], Makoto Odaka[3], Toshiaki Morikawa[3], Katsutoshi Nakayama[1], Takeo Iwamoto[4], Hirotaka Imai[2] & Kazuyoshi Kuwano[1]

Ferroptosis is a necrotic form of regulated cell death (RCD) mediated by phospholipid peroxidation in association with free iron-mediated Fenton reactions. Disrupted iron homeostasis resulting in excessive oxidative stress has been implicated in the pathogenesis of chronic obstructive pulmonary disease (COPD). Here, we demonstrate the involvement of ferroptosis in COPD pathogenesis. Our in vivo and in vitro models show labile iron accumulation and enhanced lipid peroxidation with concomitant non-apoptotic cell death during cigarette smoke (CS) exposure, which are negatively regulated by GPx4 activity. Treatment with deferoxamine and ferrostatin-1, in addition to GPx4 knockdown, illuminate the role of ferroptosis in CS-treated lung epithelial cells. NCOA4-mediated ferritin selective autophagy (ferritinophagy) is initiated during ferritin degradation in response to CS treatment. CS exposure models, using both GPx4-deficient and overexpressing mice, clarify the pivotal role of GPx4-regulated cell death during COPD. These findings support a role for cigarette smoke-induced ferroptosis in the pathogenesis of COPD.

---

[1] Division of Respiratory Diseases, Department of Internal Medicine, Jikei University School of Medicine, 105-8461 Tokyo, Japan. [2] Laboratory of Hygienic Chemistry and Medicinal Research Laboratories, School of Pharmaceutical Sciences, Kitasato University, 108-8641 Tokyo, Japan. [3] Division of Chest Diseases, Department of Surgery, Jikei University School of Medicine, 105-8461 Tokyo, Japan. [4] Division of Molecular Cell Biology, Core Research Facilities for Basic Science, Jikei University School of Medicine, 105-8461 Tokyo, Japan. [5] These authors contributed equally: Masahiro Yoshida, Shunsuke Minagawa. Correspondence and requests for materials should be addressed to S.M. (email: shunske@jikei.ac.jp)

Regulated cell death (RCD), such as that seen in apoptosis, has an essential role during a variety of physiological processes, including organogenesis and maintenance of homeostasis. Aberrant regulation of RCD has been implicated in the pathogenesis and development of many human diseases[1], and accumulating evidence now shows a role for several necrotic forms of RCD, including necroptosis, NETosis, and parthanatos, in disease pathogenesis[2–4]. Unlike apoptosis where the integrity of the plasma membrane is conserved, necrotic RCD is characterized by plasma membrane rupture and release of damage-associated molecular patterns (DAMPs) from dying cells, which can be linked to enhanced inflammatory response and tissue damage in terms of disease exaggeration[5].

Among RCDs, ferroptosis, a novel type of necrotic RCD, has recently been identified in a subset of tumor cell types expressing oncogenic RAS, which is induced by specific small molecules known as RSL and elastin[6,7]. Ferroptosis is recognized to be an iron-dependent RCD characterized by phospholipid peroxidation of plasma membranes caused by reactive oxygen species (ROS) produced during iron-mediated Fenton reactions[6]. The lipid repair enzyme glutathione peroxidase 4 (GPx4), a member of the selenoprotein family, has been shown to act as a negative regulator of ferroptosis via directly reducing lipid hydroperoxidation[8,9]. GPx4 depletion in mice caused ferroptosis in kidney tubular cells, resulting in acute renal failure, which was efficiently prevented by ferrostatin-1(Fer-1), a specific ferroptosis inhibitor[8]. Marked protection by Fer-1 was also demonstrated in the brain of mice in a collagenase-induced hemorrhagic model via inhibition of neuronal ferroptosis[10]. Although the involvement of ferroptosis in disease pathogenesis has been shown in renal and brain injury models[5,8,10,11], participation of ferroptosis in the pathogenesis of pulmonary disorders, including chronic obstructive pulmonary disease (COPD), remains unclear.

COPD is now the fourth leading cause of morbidity and mortality worldwide and is without curative treatment[12]. An aberrant inflammatory process in response to chronic cigarette smoke (CS) exposure has been recognized to be a part of COPD pathogenesis[13,14]. CS is composed of a complex mixture of over 4500 chemicals including harmful agents like free radicals[15], suggesting that multiple RCD pathways can be responsible for cell death during CS exposure. Unsurprisingly, CS has been shown to induce not only apoptosis but also necroptosis and subsequent DAMPs release from airway epithelial cells, which was causally linked to airway inflammation[16–18]. Particulate matter, including iron, is deposited in smokers' lung and can affect oxidative stress and inflammation by altering iron homeostasis[19]. Intracellular iron is mostly stored in a nonbioavailable composition by binding to ferritin, yet free iron is required for oxidative modifications to occur during Fenton reactions. Thus, disruption of iron homeostasis can elicit increased oxidative stress and tissue damage in COPD. Recent reports have revealed that labile iron is produced by autophagic ferritin degradation in a process termed ferritinophagy[20]. This new form of autophagy depends on a selective cargo receptor NCOA4 (nuclear receptor coactivator 4) that traffics ferritin to the autophagosome[21]. Importantly, ferritinophagy has been reported to promote ferroptosis[22–24]. Therefore, we hypothesized that CS induces NCOA4-mediated ferritinophagy, resulting in an increase in free iron and contributing to an increase in ferroptosis in lung epithelial cells.

Here, we show that CS promotes labile iron accumulation via NCOA4-mediated ferritinophagy, leading to phospholipid peroxidation and ferroptosis in human lung epithelial cells. Using genetic manipulation, including GPx4 transgenic mice and heterozygous GPx4-deficient mice, we verify the crucial role of ferroptosis in cigarette smoke-induced mouse models of COPD.

## Results

### CSE induces ferroptosis in Human bronchial epithelial cells (HBECs).

Free-iron-mediated lipid peroxidation is an essential process for conducting ferroptosis. To elucidate the involvement of ferroptosis in CSE-mediated cell death, CS-induced lipid peroxidation was evaluated by means of C11BODIPY staining[25] after 24 h CSE treatment in HBECs. C11BODIPY staining showed obvious lipid peroxidation in response to CSE exposure (Fig. 1a, c, f). Deferoxamine (DFO), an extracellular iron chelator, is known to be a representative inhibitor for ferroptosis. 1 h pretreatment with DFO significantly reduced the labile iron pool in HBECs, irrespective of CSE treatment, suggesting DFO enters HBECs and chelates intracellular iron in our in vitro experimental condition (Supplementary Fig. 10a). DFO clearly abrogated CSE-induced lipid peroxidation (Fig. 1a). DFO significantly inhibited CSE-induced cell death as measured by LDH and MTT assay in HBECs, respectively (Fig. 1b). Fer-1, a specific ferroptosis inhibitor, also significantly attenuated CSE-induced lipid peroxidation and CSE-induced cell death (Fig. 1c, d). We demonstrated that liproxstatin-1, a potent ferroptosis inhibitor, also prevented CSE-induced cell death (Supplementary Fig. 10b, c). Involvement of ferroptosis in CSE-induced cell death was further supported by the negligible effects on CSE-induced cell death of the necroptosis inhibitor necrostatin-1 (Nec-1) and the pan-caspase inhibitor zVAD-FMK treatment (Fig. 1d).

Next, siRNA-mediated knockdown of GPx4, an intrinsic negative regulator for lipid peroxidation and ferroptosis, was performed and efficient knockdown was observed by western blotting (Fig. 1e). GPx4 knockdown significantly enhanced CSE-induced lipid peroxidation in HBECs (Fig. 1f) and A549 cells (Supplementary Fig. 1a). GPx4 Knockdown significantly enhanced CSE-induced cell death in HBECs (Fig. 1g) and A549 cells (Supplementary Fig. 1b, c). These data suggest that ferroptosis is predominantly responsible for CSE-induced cell death in our experimental conditions.

Morphological features of ferroptosis in transmission electron microscopy (TEM) have been represented by dense and smaller mitochondria with increased membrane density and vestigial cristae[6,10]. To confirm these morphological features, TEM evaluation was performed in CSE-exposed HBECs. Dense and shrunken mitochondria were notably apparent in CSE treated HBECs compared with saline-treated HBECs (Fig. 1h). CSE-induced increase in mitochondrial membrane density was more prominent in GPx4 knockdown compared to control (Fig. 1h).

### Ferritinophagy is involved in CS-induced cell death.

Autophagic degradation of ferritin has been demonstrated to accumulate labile iron and promote ferroptosis[22,24]. Hence, to elucidate the molecular mechanisms for free iron accumulation linked to ferroptosis induction in COPD pathogenesis, we focused on NCOA4-mediated ferritinophagy in a CSE-treated bronchial epithelial cell. A recent study demonstrated that intracellular ferritin is transiently increased by CS, but declines after long time CS exposure, reflecting iron mobilization from ferritin in A549 cells[26]. In agreement with this observation, CSE treatment transiently increased ferritin expression, which was shifted to a time-dependent attenuation at 6 h with concomitant upregulation of NCOA4 in BEAS-2B (Fig. 2a, b) and HBECs (Supplementary Fig. 10d). CSE led to a time-dependent increase in IRP-2, transferrin, and ferroportin expression (Supplementary Fig. 10d). Calcein-AM assay confirmed increased free iron concentration following 5%CSE treatment, which was decreased by NCOA4 knockdown (Fig. 2c). We also confirmed the increase in free iron using inductively coupled plasma mass spectrometry (ICP-MS)[27] (Supplementary Fig. 2a). Accumulation of intracellular ferritin by

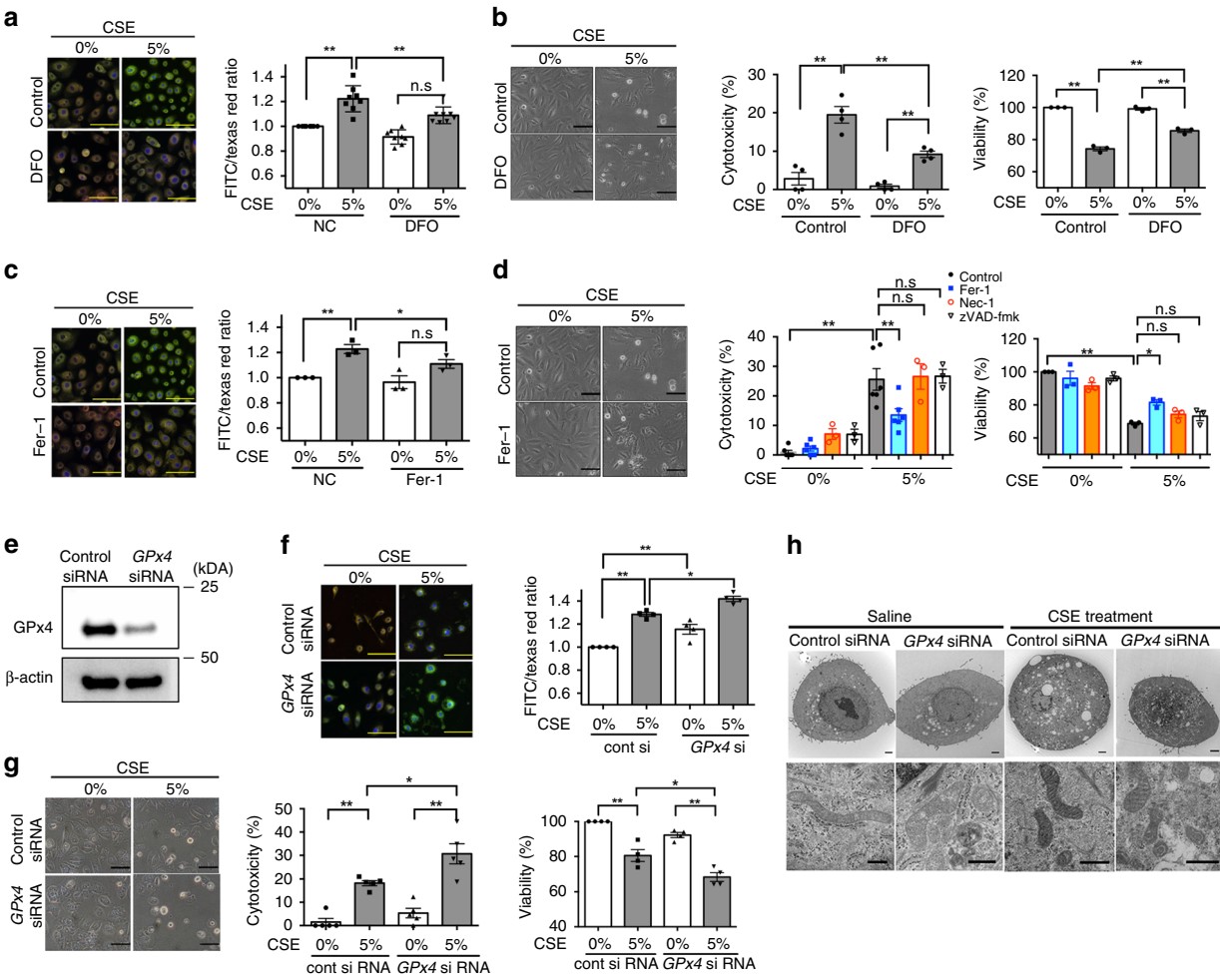

**Fig. 1** CSE induces ferroptosis in Human bronchial epithelial cells (HBECs). HBECs were treated with 5% Cigarette smoke extract (CSE) for 24 h. Lipid peroxidation assay in HBECs using the lipophilic redox-sensitive dye BODIPY 581/591, which shifts its fluorescence from red to green in response to oxidation. Scale bar = 100 µm.Cell death was assessed by cytotoxicity using LDH assay and cell viability using MTT assay. **a**, **b** Saline or Deferoxamine (DFO) (100 µM) was added to HBECs 1 hr before control or 5% CSE treatment. **a** Representative phase-contrast microscopy images of BODIPY581/591 staining are shown. Scale bar = 100 µm. n = 8 independent experiments. **b** Representative phase-contrast microscopy images of HBECs, LDH assay (n = 4 independent experiments), and MTT assay (n = 3 independent experiments) are shown. **c** Saline or Ferrostatin-1 (10 µM) was added to HBECs 1 h before control or 5% CSE treatment. Representative phase-contrast microscopy images of BODIPY581/591 staining are shown. n = 3 independent experiments. **d** Saline or Ferrostatin-1 (10 µM) or Necrostatin-1 (50 µM) or zVAD-fmk (20 µM) was added to HBECs 1 h before control or 5% CSE treatment. Representative phase-contrast microscopy images of HBECs, LDH assay, and MTT assay are shown. **e** Western blotting (WB) showing expression levels of GPx4 and β-actin in HBECs transfected with control siRNA or GPx4 siRNA. **f**, **g**. HBECs were transfected with control siRNA or GPx4 siRNA 48 h before control or 5% CSE treatment. **f** Representative phase-contrast microscopy images of BODIPY581/591staining are shown. Scale bar = 100 µm. n = 4 in each group. **g**. Representative phase-contrast microscopy images of HBECs, LDH assay, and MTT assay are shown. **a**, **c**, **f** Each panel shown represents the mean ± SEM of FITC/Texas Red ratio, taken from at least 3 independent experiments. *P < 0.05, **P < 0.01 by one-way ANOVA followed by Tukey's multiple comparisons test. **b**, **d**, **g** Each panel shown represents the mean ± SEM taken from at least 3 independent experiments. *P < 0.05, **P < 0.01 by one-way ANOVA followed by Tukey's multiple comparisons test. **h** Representative transmission electron microscopy(TEM) images of human bronchial epithelial cells transfected with control siRNA or GPx4 siRNA (top row). HBECs were treated with saline or 5% CSE for 24 h. Mitochondria are shown enlarged in bottom rows. Scale bars; 2 µm (top row), 500 nm (bottom row)

CSE treatment was enhanced by NCOA4 knockdown in bronchial epithelial cells, further indicating the participation of ferritinophagy in ferritin degradation during CSE exposure (Fig. 2d). Ferritin accumulation was also enhanced when autophagy was inhibited by ATG5 siRNA transfection (Fig. 2e) and by treatment with lysosomal protease inhibitor Pepstatin A and E64 (Fig. 2f). No significant increase in ferritin was observed by treatment with the proteasome inhibitor MG132, suggesting that the proteasome is not involved in ferritin degradation during CSE exposure (Fig. 2g). Immunofluorescence staining of BEAS-2B showed that both ferritin and LC3-GFP expression were enhanced by CSE treatment and colocalization of ferritin with LC3-GFP was detected (Fig. 2h).

Furthermore, NCOA4 knockdown in BEAS-2B reduced CSE-induced lipid peroxidation (C11BODIPY staining) and cell death (LDH and MTT assay), indicating that ferritin degradation via NCOA4-mediated ferritinophagy is responsible for ferroptosis in bronchial cell death during CS exposure (Fig. 2i–k). NCOA4 knockdown in HBECs also prevented CSE-induced cell death by means of LDH and MTT assay (Supplementary Fig. 10e, f).

Because both GPx4 and NCOA4 can modulate ferroptosis in response to CSE exposure, we examined the functional association between GPx4 and NCOA4 in regulating ferroptosis by using in vitro models. GPx4 knockdown-induced enhancement of lipid peroxidation and ferroptosis during CSE exposure was clearly

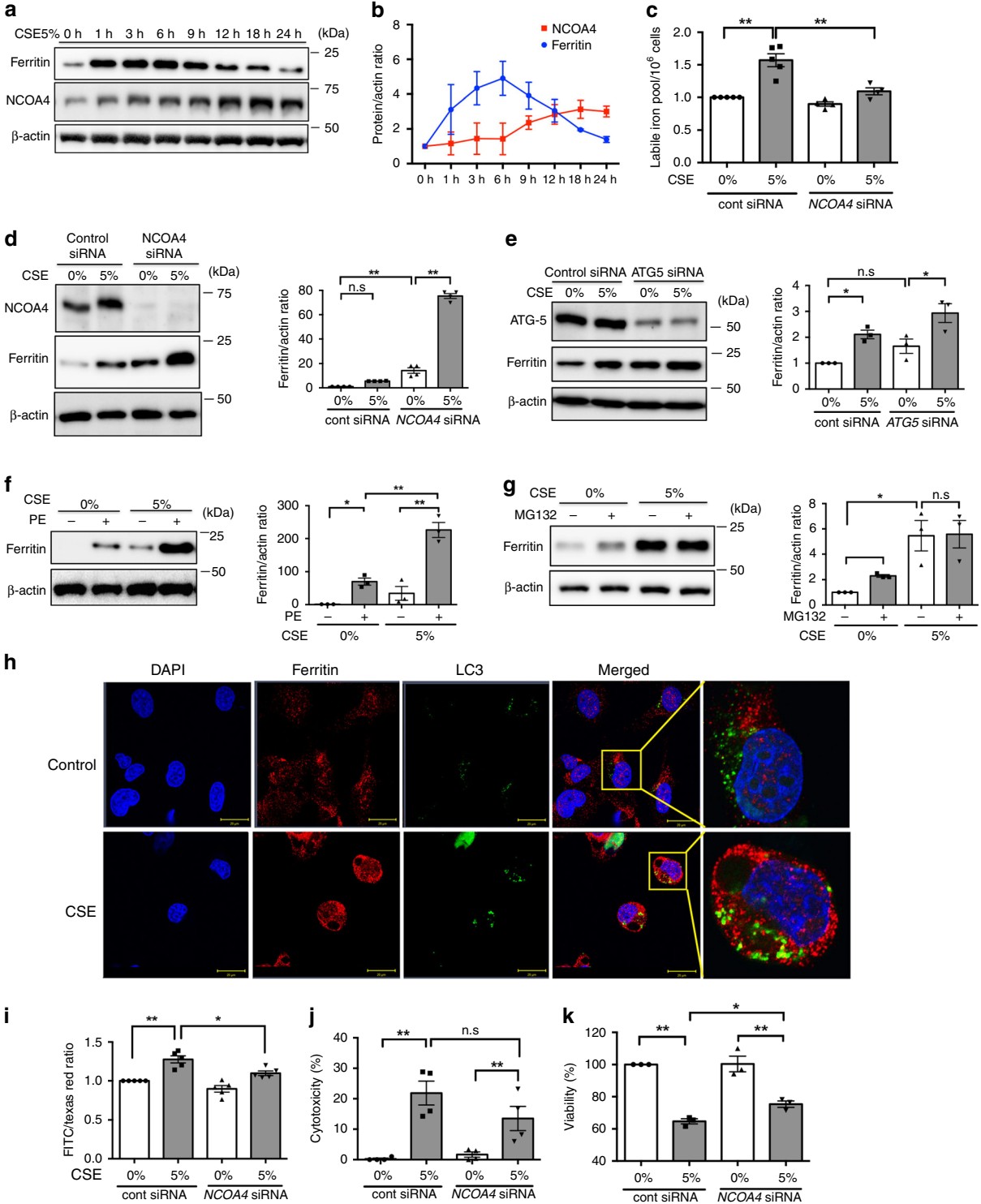

**Fig. 2** Ferritinophagy is involved in CS-induced cell death. BEAS-2B cells were treated with 5% CSE for 24 h. **a** Time course of ferritin, NCOA4, and β-actin expression in response to 5% CSE were assessed by western blotting (WB). **b** Data shown represent the mean ± SD ($n = 3$ independent experiments). **c** Labile iron pool was measured using calcein-AM method in control or 5% CSE treated BEAS-2B. Cells were transfected with control siRNA or NCOA4 siRNA 48 h before control or CSE treatment. control siRNA ($n = 5$ biologically independent samples). NCOA4 siRNA ($n = 4$ independent experiments). **d** WB showing expression levels of NCOA4, ferritin, and β-actin in control or 5% CSE treated BEAS-2B. Cells were transfected with control siRNA or NCOA4 siRNA 48 h before control or CSE treatment. ($n = 4$ independent experiments.) **e**–**g** WB showing expression levels of ferritin and β-actin in control or 5% CSE treated BEAS-2B. ($n = 3$ independent experiments.) **e** BEAS-2B cells were transfected with control siRNA or ATG5 siRNA. **f** BEAS-2B cells were treated with control or PE (Pepstatin A 10 μg/ml and E64 10 μg/ml). **g** BEAS-2B cells were treated with control or MG132. **h** Representative images of BEAS-2B show colocalization (yellow) of ferritin (red) with LC3-GFP (green). BEAS-2B were treated with 5% CSE for 6 h. Scale bars = 20 μm. **i**–**k** BEAS-2B were transfected with control siRNA or NCOA4 siRNA 48 h before control or CSE treatment. **i** BODIPY staining ($n = 5$ independent experiments), **j** LDH assay ($n = 4$ independent experiments) and **k** MTT assay ($n = 3$ independent experiments) are shown. All bar charts shown represent the mean ± SEM *$P < 0.05$, **$P < 0.01$ by one-way ANOVA followed by Tukey's multiple comparisons test

attenuated by simultaneous NCOA4 knockdown in HBECs (Supplementary Fig. 10j, k).

**CS induces ferroptosis in mouse models.** Next, we assessed whether CS increases intracellular labile iron in experimental mouse models of COPD. Wild Type mice were exposed to whole body mainstream CS for 6 months. Room air exposed WT mice served as nonsmoking controls. Perls' DAB staining disclosed that bronchial epithelial cells in CS-exposed mice had higher levels of non-heme iron compared to those in RA-exposed mice (Fig. 3a). Increased free iron, including ferric ($Fe^{3+}$) and ferrous ($Fe^{2+}$) iron, play a critical role in Fenton reactions during ferroptosis. Consistent with in vivo experiments, free iron levels in airway and whole lung homogenates were significantly increased in CS-exposed mice as measured by ICP-MS (Fig. 3b). Iron assay kit also showed significantly higher concentration of total and ferrous iron in lung homogenates from CS exposed lungs (Supplementary Fig. 3a). Our in vitro study demonstrated that intracellular ferritin was elevated transiently, but at 6 h intracellular ferritin started to decrease in a time dependent manner. In vivo experiments demonstrated that ferritin expression in CS exposed lung homogenates was significantly lower than in RA exposed lung homogenates (Fig. 3c). In line with observations in CSE-treated BEAS-2B and HBECs, NCOA4 expression levels were significantly increased in lung homogenates from CS-exposed mice (Fig. 3d).

To clarify the involvement of ferroptosis in physiologically relevant conditions, we examined mouse models using heterozygous GPx4-deficient (GPx4+/−) mice[9], GPx4 TG (TG(loxP-GPx4):GPx4+/+) mice[28], and WT(GPx4+/+) mice as previously described[29]. Both GPx4+/− and GPx4 TG mice grow normally and GPx4 protein expression levels in lung homogenates were corresponding to their genetic status (Supplementary Fig. 3b). After CS exposure for 6 months, lipid peroxidation was evaluated by means of liquid chromatography-mass spectrometry (LC-MS) and 4-HNE expression, respectively. LC-MS/MS showed that CS exposure weakly increased phosphatidylcholine hydroperoxide (PC-OOH)/ phosphatidylcholine (PC) ratios and phosphatidylethanolamine hydroperoxide (PE-OOH)/ phosphatidylethanolamine (PE) in wild type mice, which was markedly enhanced in GPx4+/− mice (Fig. 3e, f, Supplementary Fig. 4,5). Consistent with PC peroxidation, 4-HNE staining in wild type mice was increased by CS exposure, which was enhanced in GPx4+/− mice but reduced in GPx4 TG mice in both airway epithelial cells (Fig. 3g) and alveolar epithelial cells (Fig. 3g). Western blotting for 4-HNE expression levels in lung homogenates also showed a similar trend (Supplementary Fig. 3c).

Next, cell death in mouse lung was evaluated by a combination of TUNEL assay and cleaved caspase-3 expression. TUNEL positive cells were slightly but significantly increased in CS exposed WT mice lung, were significantly enhanced in GPx4+/− mice relative to WT, and were significantly reduced in GPx4 TG mice relative to WT (Fig. 3h). Based on DAPI staining (Supplementary Fig. 3e), we speculated that the TUNEL positive cells were mainly comprised of airway epithelial cells and alveolar epithelial cells. Only a small number of cleaved caspase-3 positively stained apoptotic cells were detected in CS exposed mice lungs, but no difference relative to GPx4 genetic status was apparent (Fig. 3i). Cleaved caspase-3 protein levels in lung homogenates were also slightly increased by CS exposure, but no alteration was detected in GPx4+/− and GPx4 TG mice (Fig. 3j), indicating that significantly increased cell death in CS-exposed GPx4+/− mice is mainly attributable to ferroptosis.

Although GPx4-regulated lipid peroxidation was obviously involved in the regulation of ferroptosis in CS-exposed mouse models, no significant difference of NCOA4 and ferritin expression levels was demonstrated based on the alterations of GPx4 status (Supplementary Fig. 3d), suggesting the absence of direct link between GPx4 and NCOA4 in terms of protein expression levels. Hence, to confirm the involvement of NCOA4-mediated ferritinophagy in ferroptosis during CS exposure, we performed an in vivo CS experiment using NCOA4 siRNA. WT mice were exposed to whole body mainstream CS for 7days. Control siRNA or NCOA4 siRNA was injected intra-tracheally by using in vivo-jetPEI on day1. IHC staining demonstrated that NCOA4 siRNA injection efficiently diminished NCOA4 expression levels in mouse bronchial epithelial cells (BEC) (Supplementary Fig. 6a). To evaluate ferritin and 4-HNE expression levels in BEC, consecutive sections of mouse lung tissues were used. Ferritin expression was enhanced in NCOA4 siRNA-treated BEC compared to control siRNA treated BEC, suggesting NCOA4 reduction was associated attenuated ferritin degradation of ferritinophagy in BEC (Supplementary Fig. 6a). 4-HNE expression was also significantly decreased in NCOA4 siRNA treated BEC, indicating attenuated lipid peroxidation can be attributed to less ferritin degradation with concomitantly reduced Fenton type reactions in the setting of NCOA4 reduction (Supplementary Fig. 6a). Next, dead cells were counted in mouse lung by means of TUNEL assay. TUNEL positive cells were significantly increased by CS exposure, which were clearly decreased by NCOA4 knockdown (Supplementary Fig. 6b). Taken together, these results indicate that both GPx4-regulated lipid peroxidation and NCOA4-mediated ferritinophagy are involved in CS-induced ferroptosis in mouse models.

**GPx4 modulates COPD phenotype in smoking mouse models.** Necrotic cell death, including ferroptosis, can amplify inflammation via release of DAMPs to the extracellular environment. Hence, we counted inflammatory cells obtained in bronchoalveolar lavage fluid (BALF) from the lungs of GPx4-TG mice, WT (GPx4+/+) mice, and GPx4+/− mice following 4weeks of CS exposure. Total cell and macrophage counts in BALF were significantly increased in response to CS exposure in wild type mice. Compared to wild type mice, a further increase in total cell, macrophage, and lymphocyte counts in BALF was demonstrated in GPx4+/− mice (Fig. 4a). In contrast, no significant increase in cell counts were observed in BALF from GPx4 TG mice (Fig. 4a). DAMPs, including IL-33 and IL-1α in lung homogenates were significantly increased by CS exposure in wild type mice and were further enhanced in GPx4+/− mice (Fig. 4b, c). Compared to CS-exposed GPx4+/− type mice, significant reduction of HMGB-1 in BALF was detected in CS-exposed GPx4 TG mice (Fig. 4d). Furthermore, expression of the proinflammatory cytokine TNF-α was elevated. CS-induced upregulation of TNF-α was inhibited in GPx4 TG mice (Fig. 4e). Collectively, CS exposure induces ferroptosis accompanied by release of DAMPs and proinflammatory cytokines from lung epithelial cells.

To examine the involvement of GPx4-regulated ferroptosis in COPD development, morphometric analysis of airspace enlargement and small airway thickness after 6 months of CS exposure were performed in GPx4+/−, GPx4 TG, and WT mice with corresponding room air controls. CS exposure induced significant enlargement in lung airspace mimicking emphysematous changes in WT mice as measured by alveolar mean linear intercept (Lm) (Fig. 4f). CS-induced lung airspace enlargement was significantly enhanced in GPx4+/− mice but obviously attenuated in GPx4 TG mice (Fig. 4f). CS-exposed WT mice showed increase in airway wall thickening compared to air-treated control mice, which was significantly enhanced in GPx4+/− mice, but no alteration was detected in GPx4 TG mice (Fig. 4g).

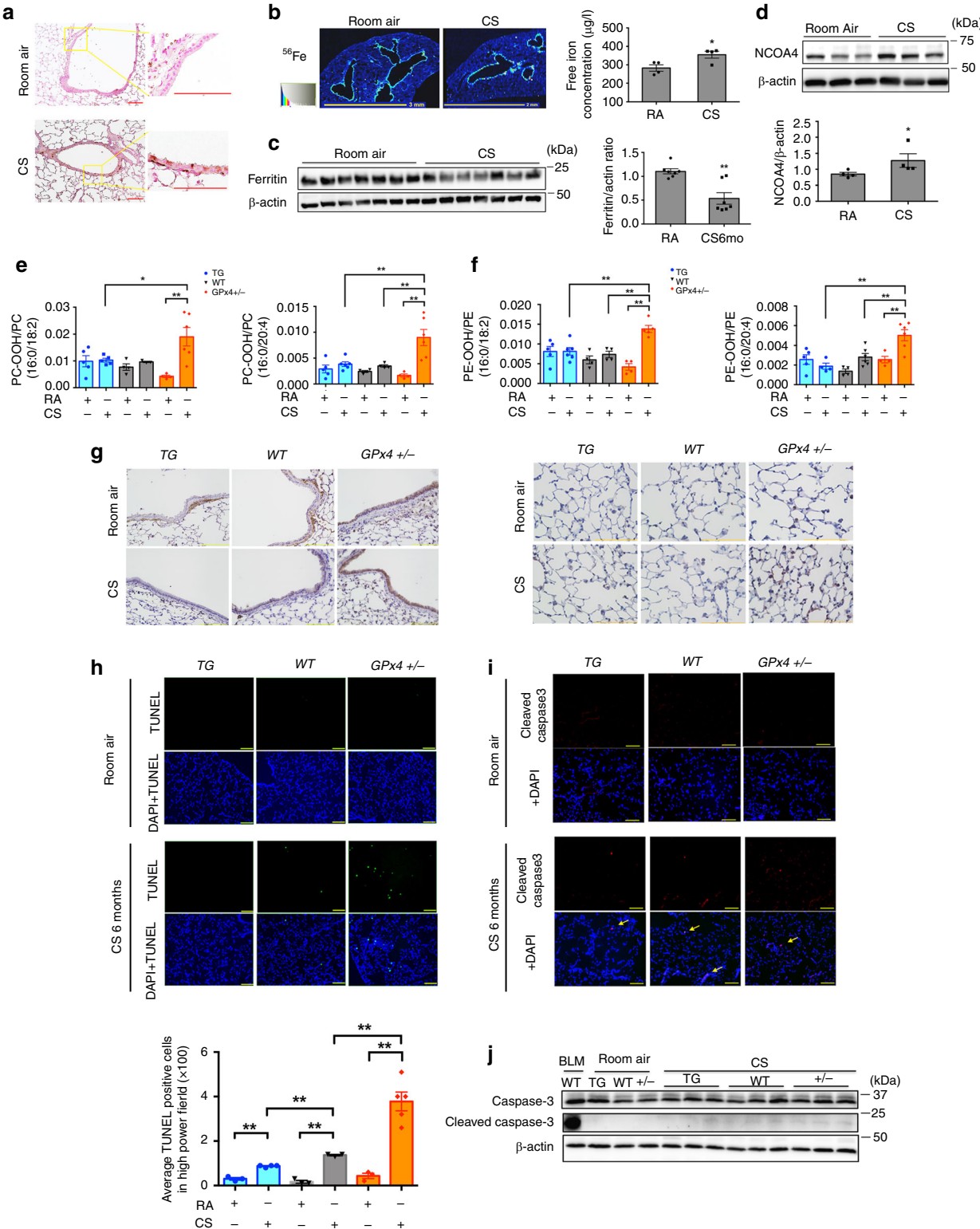

Next, TEM of bronchial epithelial cells was examined in WT and *GPx4*+/− mice. Accumulation of mitochondria with increased membrane density was demonstrated in CS exposed WT mice, which was barely detected in air-treated control mice (Fig. 4h). Intriguingly, accumulation of mitochondria with increased membrane density accompanied by disrupted cristae was a prominent feature in CS-exposed *GPx4*+/− mice (Fig. 4h).

**Involvement of ferroptosis in human COPD lung**. To elucidate the potential involvement of GPx4 in COPD pathogenesis, we evaluated GPx4 expression levels in human bronchial epithelial cells isolated from never smoker ($n = 6$), smoker non-COPD ($n = 6$), and COPD patient lung ($n = 7$). Western blotting demonstrated that GPx4 expression was significantly lower in HBECs from COPD lungs than in never smoker's and non COPD smoker's lungs (Fig. 5a). Furthermore, GPx4 expression levels

**Fig. 3** CS induces ferroptosis in mouse models. **a, c, d** WT mice were exposed to Room Air (RA) or cigarette smoke (CS) for 6 months. Ferric iron deposits were stained with Perls' DAB staining in lung samples from RA (upper) or CS (lower) exposed mice. Counter staining is Fast red. Image (original magnification, ×400) is representative of 5 images/mouse. 3 biologically independent mice/group were analyzed. Scale bar = 100 μm. **b** The amount of non-heme iron was measured by means of inductively coupled plasma mass spectrometry (ICP-MS) ($n = 4$ biologically independent mice). *$P < 0.05$ by student's $t$-test. **c** WB showing expression levels of ferritin and β-actin in lung homogenate of WT mice. *$P < 0.05$ by student's $t$-test. ($n = 7$ biologically independent mice). **d** WB showing expression levels of NCOA4 and β-actin in lung homogenates from RA ($n = 4$) or CS ($n = 4$) exposed mice. ($n =$ biologically independent mice.) *$P < 0.05$ by student's $t$-test. **e–j** Heterozygous GPx4-deficient mice (GPx4+/−), GPx4+/+ (WT), and TG(loxP-GPx4): GPx4+/+ (GPx4TG) mice were exposed to RA or CS for 4 weeks. **e** LC-MS analysis of PC and their oxidation products in lung homogenates. PC-OOH/PC (16:0/18:2), (16:0/20:4) were shown. **f** LC-MS analysis of PE and their oxidation products in lung homogenates. PE-OOH/PE (16:0/18:2), (16:0/20:4) were shown. **e, f** TG RA $n = 5$,TG CS $n = 6$, WT RA $n = 4$,WT CS $n = 4$, +/− RA $n = 4$, +/− CS $n = 6$. ($n =$ biologically independent mice) *$P < 0.05$, **$P < 0.01$ by one-way ANOVA followed by Tukey's multiple comparisons test. **g** Immunohistochemical(IHC) staining of 4-HNE in mouse lung airway (upper panels) and parenchyma (lower panels). Original magnification ×400. Bar = 100 μm. **h** TUNEL assay staining (green) in GPx4 +/−, WT, and GPx4 TG mice lung sections. Nuclei were counterstained with DAPI (blue). TG RA $n = 3$,TG CS $n = 4$, WT RA n = 3,WT CS $n = 3$, +/− RA n = 3, +/− CS $n = 5$. ($n =$ biologically independent mice)**$P < 0.01$ by one-way ANOVA followed by Tukey's multiple comparisons test. **i** Immunofluorescence staining of cleaved caspase3 (red). Nuclei were counterstained with DAPI (blue). Original magnification ×200. Bar = 100 μm. **j** WB showing expression levels of caspase 3, cleaved caspase3, and β-actin in lung homogenate of GPx4+/−, WT, and GPx4 TG mice. All bar charts shown represent the mean ± SEM

were positively correlated with percentage of FEV1/FVC (Fig. 5a). Immunohistochemistry of lung tissues also showed lower expression of GPx4 in bronchial epithelial cells in COPD patients than in non COPD patients (Fig. 5b). GSH/GSSG (glutathione) ratio and total glutathione level were decreased in response to CSE exposure in HBEC for 24 h. In addition, GSH was clearly upregulated concomitantly with suppression of lipid peroxidation by N-acetylcysteine (Supplementary Fig. 10g, h, i). These results indicate that decline of GPx4 and its substrate GSH can be linked to an insufficient anti-oxidant stress response to CS, resulting in induction of ferroptosis.

Next, participation of lipid peroxidation, NCOA4-mediated ferritinophagy, and ferroptosis was examined using human lung samples. NCOA4 expression levels were significantly increased in lung homogenates from COPD patients in comparison to never smokers and non-COPD smokers (Fig. 5c). Intriguingly, correlation analysis in human lung homogenates from never smokers ($n = 10$), non-COPD smokers ($n = 6$), and COPD patients ($n = 15$) revealed that NCOA4 protein level was negatively associated with percentage of forced expiratory volume in one second (FEV1)/forced vital capacity (FVC) (Fig. 5c). Immunohistochemistry of lung tissue also demonstrated enhanced expression of NCOA4 in bronchial epithelial cells, especially in COPD patients (Fig. 5d). To clarify the causal link between excessive iron levels and lipid peroxidation, consecutive sections of human lung tissues were used for perls' DAB staining and for 4-HNE detection. In contrast to never smoker's lungs and non COPD smoker's lungs, obvious positive 4-HNE staining was detected in COPD lung, predominantly in the cytoplasm of bronchial epithelial cells with concomitantly enhanced perls' DAB staining (Fig. 5f). Western blotting in lung homogenates showed a similar tendency for 4-HNE expression levels, but large variation for ferritin expression (Supplementary Fig. 7a,b). However, lung homogenate contains wide variety of cell types including inflammatory cells. In the present study, we clearly demonstrated increased free iron in lung epithelial cells (Fig. 2c, Fig. 3a, b, Fig. 5f). Lipid oxidation products were analyzed using LC-MS in human lung homogenates from non-smokers ($n = 9$), non-COPD smokers ($n = 7$), and COPD patients ($n = 10$). PC-OOH/PC ratios and PE-OOH/PE in COPD lung homogenates were significantly higher than those in non-COPD lung homogenates (Fig. 5e, Supplementary Fig. 8,9). TEM evaluation was also performed in human lung samples. In good agreement with recent reports associated with ferroptosis[6,10], accumulation of small sized mitochondria with increased membrane density was obvious in airway epithelial cells in COPD lung, but was barely detected in airway epithelial cells in non-smoker lungs (Fig. 5g).

## Discussion

In this study, we demonstrate the likely involvement of GPx4-regulated ferroptosis associated with NCOA4-mediated ferritinophagy in COPD pathogenesis. Our in vitro models using DFO, Fer-1, and GPx4 knockdown clarified the involvement of ferroptosis in CSE-induced cell death in airway epithelial cells. Increased NCOA4, a selective cargo receptor for autophagic recognition of ferritin, was linked to ferritinophagy and subsequent release of free iron causally linked to lipid peroxidation during CS exposure. In in vivo CS-exposed mouse models, GPx4+/− mice showed remarkably higher levels of lipid peroxidation, non-apoptotic cell death, DAMPs release, and enhanced COPD phenotypes of airspace enlargement and small airway thickness relative to WT mice, all of which are attenuated in GPx4 TG mice. NCOA4 knockdown attenuated lipid peroxidation and ferroptosis in response to CS exposure in WT mice. COPD lung tissues showed accumulation of iron with concomitant increase in lipid peroxidation. GPx4 expression levels in HBECs isolated from COPD lung were significantly lower than those from non COPD lung, which were positively correlated with FEV1% (Fig. 5a, b). NCOA4 expression levels in COPD lung tissues were higher than those in normal lung tissues, and intriguingly, were negatively correlated with %FEV1 (Fig. 5c), suggesting a link between ferritinophagy and worsening of airway obstruction in the setting of reduced GPX4 and increased NCOA4 during COPD pathogenesis. TEM evaluation further confirmed the existence of ferroptosis in airway epithelial cells in response to CS exposure and in COPD lungs. Taken together, these findings suggest that alteration of iron homeostasis by CS-induced ferritinophagy and subsequent induction of ferroptosis play a novel role in COPD pathogenesis.

Regulated cell death, including apoptosis and necroptosis, in lung epithelial cells has been shown to have an important role in COPD pathogenesis. Apoptosis in both alveolar and airway epithelial cells and T-lymphocytes were demonstrated in COPD lungs in association with tissue damage and inflammation[30]. Involvement of necroptosis and increased expression levels of its regulatory molecule RIP3 in lung epithelial cells were demonstrated in CS-exposed mouse models and human COPD lungs[16]. However, neither zVAD-fmk, an apoptosis inhibitor, nor necrostatin-1, a necroptosis inhibitor, showed apparent inhibition of CSE-induced cell death in HBECs in our in vitro experimental conditions. Different doses of cigarette smoke extract may elicit different types of cell death at different time points and so it is not as clear cut to say one type of death is predominant. However in the present study, we clearly show that CSE-induced cell death is at least partly attributable to ferroptosis in HBECs. Ferroptosis

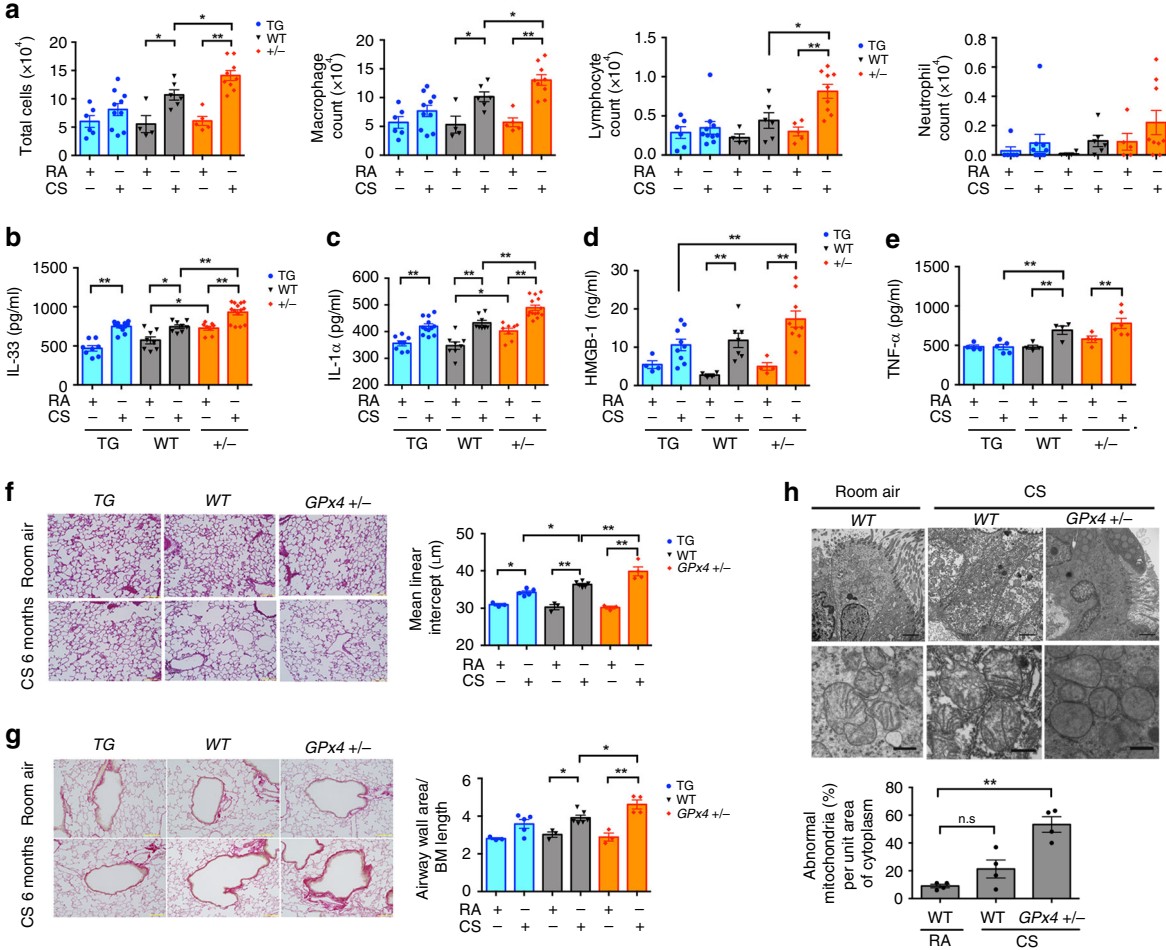

**Fig. 4** GPx4 modulates COPD phenotype in smoking mouse models. **a–e** GPx4+/−, GPx4+/+ (WT), and TG(loxP-GPx4):GPx4+/+ (GPx4TG) mice were exposed to RA or CS for 4weeks. **a** Cell counts of total cells, macrophages, lymphocytes, neutrophils in bronchoalveolar lavage fluid (BALF) from RA or CS exposed mice. $n = 6$ (mice) in TG-RA(TR), $n = 10$ TG-CS(TC), $n = 4$ in WT-RA(WR), $n = 6$ in WT-CS(WC), $n = 5$ in GPx4+/− RA (GR), $n = 9$ in GPx4+/− CS(GC). **b** ELISA assay of IL-33 ($n = 8$ in TR, $n = 12$ TC, $n = 8$ in WR, $n = 8$ in WC $n = 8$ in GR, $n = 14$ in GC), **c** IL-1α ($n = 8$ in TR, $n = 12$ TC, $n = 8$ in WR, $n = 8$ in WC, $n = 8$ in GR, $n = 14$ in GC), **e** TNF-α ($n = 4$ in TR, $n = 5$ TC, $n = 4$ in WR, $n = 4$ in WC, $n = 4$ in GR, $n = 6$ in GC), **d** in lung homogenate in BALF and **d** HMGB-1 in BALF($n = 4$ in TR, $n = 9$ in TC, $n = 4$ in WR, $n = 7$ in WC, $n = 4$ in GR, $n = 9$ in GC), from RA or CS exposed mice. **f, g** Mice were exposed to RA or CS for 6 months. Lung sections were stained with haematoxylin and eosin. Representative histological images of GPx4+/−, WT, and GPx4 TG mice lung parenchyma. Original magnification ×200. Bar = 100 µm. **f** Alveolar size was estimated by means of the mean linear intercept (MLI) method. $n = 3$ in TR, $n = 5$ TC, $n = 3$ in WR, $n = 6$ in WC, $n = 3$ in GR, $n = 4$ in GC. **g** Representative histological images of GPx4+/−, WT, and GPx4 TG mice airways. Original magnification ×200. Bar = 100 µm. Alveolar wall thickness determined by calculating the airway wall/basement membrane length using Image J. $n = 3$ in TR, $n = 5$ TC, $n = 3$ in WR, $n = 6$ in WC, $n = 3$ in GR, $n = 4$ in GC **h** Representative TEM image of bronchial epithelial cell of Room Air exposed WT mice (left, top), CS exposed WT mice (middle, top), and CS exposed GPx4+/− mice (right, top). Mitochondria are shown enlarged in bottom rows. Scale bars: 2 µm (top row), 500 nm (bottom row). ($n = 4$ biologically independent mice). All data shown represent the mean ± SEM. *$P < 0.05$, **$P < 0.01$ by one-way ANOVA followed by Tukey's multiple comparisons test. Throughout, $n =$ biologically independent mice

was recently identified as a novel type of necrotic RCD characterized by free iron-dependent phospholipid peroxidation of cell membranes, which is negatively regulated by the selenoprotein GPx4[6,8]. In comparison to other types of necrosis, any characteristic alterations in structure for ferroptosis remain obscure. Recent papers showed that small-sized mitochondria with increased membrane density and vestigial cristae can be specific findings in ferroptosis[6,10]. Consistently, our TEM evaluations detected an obvious accumulation of similarly structured mitochondria in airway epithelial cells in not only GPx4 depleted experimental conditions, but also in COPD lungs (Fig. 5g), further supporting the involvement of ferroptosis in COPD development. Given reports that several other cell death pathways participate in COPD pathogenesis, it is possible that ferroptosis is intertwined with other cell death pathways. Recently, a novel and

intriguing hypothesis has emerged, which proposes that some necrotic forms of RCD appears to be interconnected with each other through DAMPs release, and form necro-inflammatory auto amplification loops, resulting in tissue damage and organ dysfunction[5]. Our cytokine analysis revealed that CS promoted release of TNF-α, a representative necroptosis trigger, as well as DAMPs release, which was regulated by GPx4 (Fig. 4b–e), suggesting that ferroptosis might trigger other form of necrotic RCDs including necroptosis as a part of pathogenic amplification loops.

GPx4 is a selenoprotein glutathione peroxidase that directly reduces peroxidized phospholipids in cell membranes for redox homeostasis[31,32]. GPx4 has a crucial role during development and whole body ablation of the GPx4 gene in mice was found to be embryonic lethal between 7.5 and 8.5 days post-coitum, with

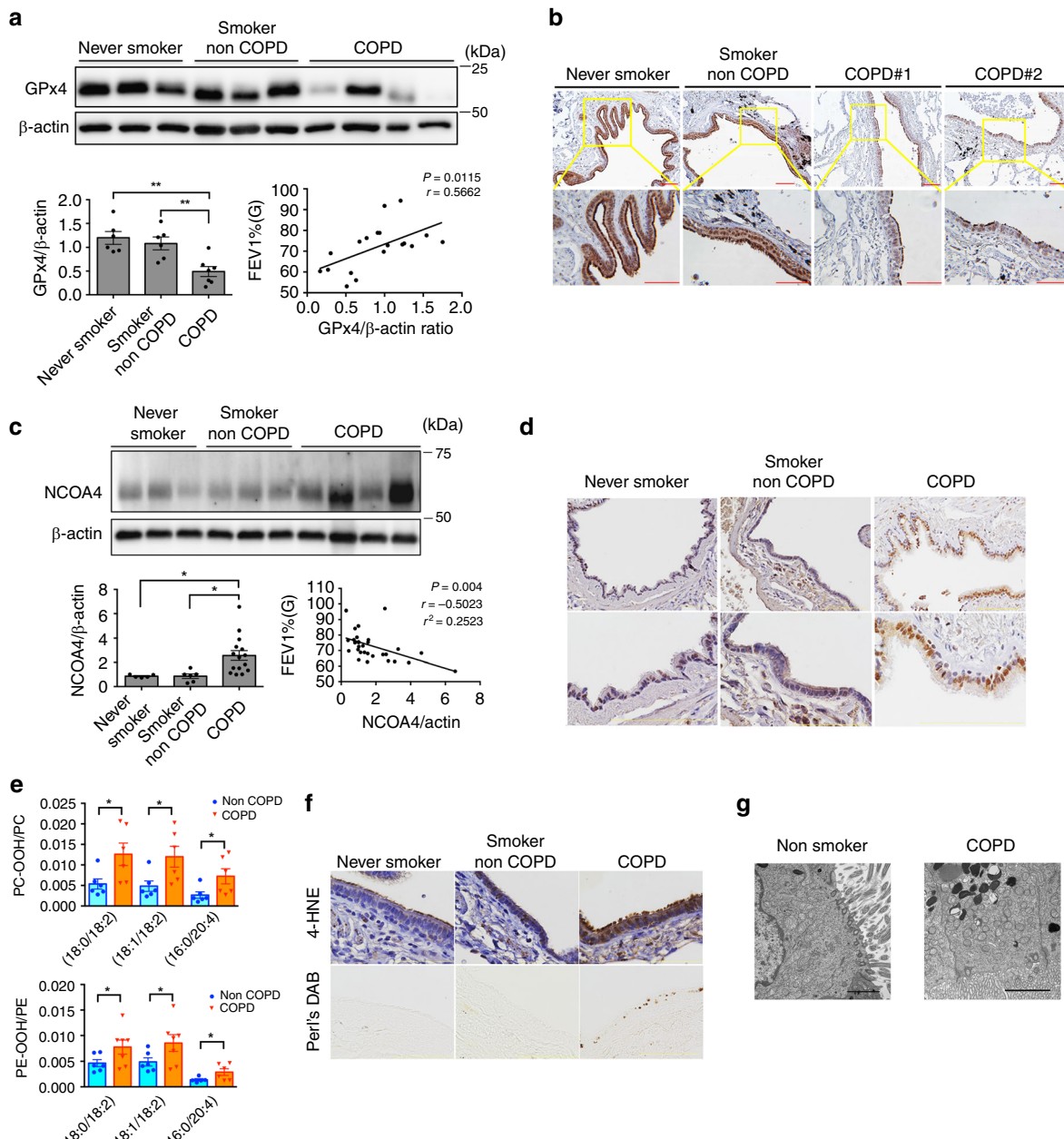

**Fig. 5** Involvement of ferroptosis in human COPD lung. **a** WB showing expression levels of GPx4 and β-actin in HBECs isolated from never smokers ($n = 6$) or non COPD smokers ($n = 6$) or COPD patients ($n = 7$). The bar chart shown represent the mean ± SEM. *$P < 0.05$, **$P < 0.01$ by one-way ANOVA followed by Tukey's multiple comparisons test. The correlation analysis between GPx4 expression level and the percentages of FEV1/FVC ($n = 19$) in HBECs from never smokers, non-COPD smokers, and COPD patients. **b** IHC of GPx4 in never smoker, healthy smoker and COPD lung. Original magnification ×400. Bar = 100 μm. **c** WB showing expression levels of NCOA4 and β-actin in lung homogenates from never smokers ($n = 5$) or non COPD smokers ($n–6$) or COPD patients ($n = 15$). The bar chart shown represent the mean ± SEM. *$P < 0.05$, **$P < 0.01$ by one-way ANOVA followed by Tukey's multiple comparisons test. The correlation analysis between NCOA4 expression level and the percentages of FEV1/FVC ($n = 31$) in human lung homogenates from never smokers ($n = 10$), non-COPD smokers ($n = 6$), and COPD patients ($n = 15$). **d** IHC of NCOA4 in human never smoker, healthy smoker and COPD lung. Original magnification ×400. Bar = 100 μm. **e** LC-MS analysis of PC and their oxidation products in never smokers, smoker non-COPD, and COPD lung homogenates. MS of precursor ion scanning of m/z 184 in the positive ion mode were analyzed. PC-OOH/PC (18:0/18:2), (18:1/18:2), (16:0/20:4) shown in upper panel. non-COPD, n = 6; COPD, n = 6. LC-MS analysis of PE and their oxidation products in never smokers, smoker non-COPD, and COPD lung homogenates. *$P < 0.05$ by student's t-test. MS of neutral loss scanning of 141 Da in the positive ion mode were analyzed. PE-OOH/PE (18:0/18:2), (18:1/18:2), (16:0/20:4) were shown in lower panel. non-COPD, n = 6; COPD, n = 7. *$P < 0.05$ by by student's t-test. **f** Perls' DAB and 4-HNE staining in never smoker, healthy smoker and COPD lung tissues (original magnification, ×400). Bar = 100 μm. **g** Representative TEM images of bronchial epithelial cells in never smoker lung and COPD lung (Brinkman Index = 2100, FEV1% = 62.5%). Scale bars ;2 μm. Throughout, n = independent patients

concomitantly enhanced cell death[9]. In addition, GPx4 knockout specifically in forebrain[33] and kidney[8] has been reported to show severe pathological phenotypes in association with ferroptosis, respectively. Hence, GPx4 has been recognized to be a central regulator of ferroptosis[32,34,35]. In this study, we used *GPx4+/−* mice[9] and *GPx4 TG* mice[28] to clearly demonstrate the involvement of ferroptosis in COPD pathogenesis. We observed reduced expression of GPx4 in HBECs isolated from COPD lungs and significant correlation between COPD severity and GPx4 expression levels in HBECs, suggesting that GPx4 modified mouse models can be physiological COPD model (Fig. 5a, b). GPx4 utilizes the reducing equivalents from its substrate GSH(gluthatione) and it has been reported that GSH levels were reduced by CSE in BEAS2B[36]. Several reports demonstrated that ferroptosis is induced by loss of GPx4 activity due to the depletion of intracellular glutathione via the anti-cancer drug Erastin-mediated inhibition of the xCT cysteine transporter[37], or by direct binding of RSL3 to selenocysteine in the GPx4 enzymatic active site[8]. Hence, we speculate that reduced GSH levels can be linked to insufficient GPx4 activity. In line with previous findings, we observed clear reduction of GSH in response to 24 h CSE exposure in HBECs (Supplementary Fig. 10g, h). Furthermore, NAC (anti-oxidant via GSH upregulation) treatment increased GSH levels and clearly reduced lipid peroxidation, further supporting the notion that not only GPx4 levels but also GSH levels are a critical determinant for regulating lipid peroxidation and subsequent ferroptosis (Supplementary Fig. 10g, h, i). Taken together, we speculate that not only reduced GPx4 expression levels but also loss of GPx4 activity play a crucial role in COPD pathogenesis.

In comparison to non-smoker lungs, BAL fluids recovered from smoker lungs contain higher concentrations of iron and ferritin[38,39]. Increased iron burden in the lower respiratory tract has been reported in association with cigarette smoking[40]. A recent paper showed that iron-responsive element-binding protein2 (IRP2) upregulates mitochondrial iron loading in association with CS-induced pulmonary injury[41]. Previous studies have delineated that cigarette smoke-induced release of iron from ferritin influences iron status during cigarette smoking[42]. Free iron-mediated Fenton reactions have been implicated in lipid peroxidation during ferroptosis[22–24], hence, it is plausible that disruption in iron homeostasis with increased free-iron may have a pivotal role in ferroptosis during COPD pathogenesis. In fact, we observed increased free-iron with concomitant lipid peroxidation in response to CSE exposure, which was mainly attributable to ferritinophagy (Figs. 1, 2).

Autophagy is a catabolic process by which intracellular substrates are degraded in lysosomes to maintain cell viability and homeostasis in response to nutrient stress, hypoxia, as well as smoking stress[43]. The involvement of autophagy and mitochondria-selective mitophagy in COPD pathogenesis have been reported in terms of cellular senescence and RCD regulation, respectively[16,44–46]. Ferritinophagy is recently identified as a novel selective autophagic degradation of ferritin mediated by the specific adaptor protein NCOA4[20,21,47]. Our in vitro and in vivo experiments showed an inverse relationship between ferritin and NCOA4 expression during CSE exposure in HBECs (Fig. 2). SiRNA-mediated knockdown experiments clarified the involvement of NCOA4 in ferritin degradation, and confocal microscopic evaluation showed colocalization between ferritin and autophagosomes, indicating a specific role for NCOA4-mediated ferritinophagy in ferritin degradation during CSE exposure (Fig. 2). NCOA4 knockdown significantly reduced lipid peroxidation and recovered cell viability in response to CSE exposure, further supporting the notion that free iron release by NCOA4-mediated ferritinophagy has a regulatory role in CSE-induced ferroptosis (Fig. 2). Intratracheal NCOA4 siRNA injection in WT mice

reduced lipid peroxidation and also reduced CS induced cell death in airway, partially supporting the notion that NCOA4-mediated ferritinophagy has a crucial role in ferroptosis (Supplementary Fig. 6). However, this in vivo siRNA experiment has potential non-specific and non-selective limitations. Further in vivo evidence such as epithelial cell specific NCOA4 knock out mice is needed to support the role of NCOA4 in the development of a COPD phenotype. Intriguingly, significant correlation between NCOA4 expression levels in lung homogenates and a decline in FEV1/FVC was demonstrated, which may reflect clinical implications for NCOA4-mediated ferritinophagy in COPD.

In the present study, direct association between GPx4 and NCOA4 has not been demonstrated in terms of protein expression levels in our mouse experiment (Supplementary Fig. 3d). However, our in vitro siRNA experiment revealed that NCOA4 knockdown attenuated exaggerated ferroptosis induced by GPx4 knockdown in CSE exposed HBECs (Supplementary Fig. 10j, k), suggesting the existence of a functional link between GPx4 and NCOA4 with respect to ferroptosis regulation. Because COPD samples clearly showed both reduced GPx4 and increased NCOA4 protein levels, it is plausible that this functional link can be important in enhancing ferroptosis during COPD pathogenesis.

In conclusion, our findings support the likely role in COPD pathogenesis of ferritinophagy-mediated ferritin degradation during ferroptosis and subsequent lipid peroxidation. Accumulation of free iron with concomitantly enhanced NCOA4-regulated ferritinophagy during CS exposure may at least partly explain the disruption of iron homeostasis in COPD lungs. We believe that this study provides important clues for developing novel COPD treatments targeting ferroptosis by regulating iron hemostasis and lipid peroxidation.

## Methods

**Cell culture, antibodies, and reagents**. Normal and COPD lung tissues were obtained from pneumonectomy and lobectomy specimens from primary lung cancer. Informed consent was obtained from all surgical participants as part of an approved ongoing research protocol by the ethical committee of Jikei University School of Medicene (#20-153 (5443)). Primary human bronchial epithelial cells (HBECs) were isolated using protease treatment. Sterilely, isolated HBECs were seeded on rat-tail collagen type I-coated (10 μg/ml) dishes,and incubated overnight. Then the medium was changed to bronchial epithelial growth medium (BEGM, Lonza, CC-3170). Cultures were characterized by >95% positive staining with anti-cytokeratin and <5% positive staining with the anti-vimentin antibody[46]. Bronchial epithelial cell line BEAS-2B (ATCC,Manassas,VA) was cultured in RPMI 1640 (Gibco Life Technologies, 11875-093) with 10% FCS (Gibco Life Technologies, 26140-079) and penicillin-streptomycin (Gibco Life Technologies, 15140-122). The alveolar type II cell-like epithelial cell line A-549 cells (ATCC,Manassas,VA) was cultured in DMEM (Gibco Life Technologies, 11965-092) with 10% FCS and penicillin-streptomycin.

The antibodies used were rabbit anti-Ferritin Heavy Chain (Abcam, 65080), rabbit anti-NCOA4 (Abcam, 86707), rabbit anti-NCOA4(Thermo Fischer Scientific,PA5-36391) rabbit anti-ATG5 (Cell Signaling Technology, 2630), rabbit anti-caspase-3 (Cell Signaling Technology, 9665), rabbit anti-cleaved caspase-3 (Cell Signaling Technology, 9664), rabbit anti-GPx4 (Abcam, 125066), rabbit anti-4-HNE (Abcam, 46545), rabbit anti-IRP-1(Cell signaling, 20272), rabbit anti-IRP-2 (Cell signaling, 37135), rabbit anti-Transferrin receptor (Cell signaling, 13113), goat anti-Ferroportin (Santa cruz,sc-49668), and mouse anti-β-actin (Sigma-Aldrich, A5316). Antibody dilutions were according to manufacturer's instructions and information for dilutions were listed in Supplementary Table 1.

The following reagents were used: Deferoxamine (Sigma-Aldrich, D9533), Ferrostatin-1 (Sigma-Aldrich, SML0583), Necrostatin-1 (Sigma-Aldrich, N9037), z-VAD-fmk (Promega, G7231), pepstatin A (Peptide Institute, 4397), E64d (Peptide Institute, 4321-v), bafilomycin A1 (Sigma-Aldrich, B1793), MG-132 (Enzo Life Science, BML-P102), Liproxstatin-1 (Sigma-Aldrich, SML1414), GSH/GSSG-Glo Assay(Promega,#V6611), N-acetylcysteine (NAC)(Wako,#017-05131), Hoechst 33258 (Sigma-Aldrich, 861405), and 16:0 PC-d75 (Avanti Polar Lipids, 860358).

**Plasmids, small interfering RNA and transfection**. The LC3B cDNA was the kind gift of Dr. Mizushima (Tokyo University, Tokyo, Japan) and Dr. Yoshimori (Osaka University, Osaka, Japan), and was cloned into *pEGFP-C1* vector. Small interfering RNA (siRNA) targeting GPx4 (Applied Biosystems Life Technologies, s6112), NCOA4 (Applied Biosystems Life Technologies, s15566, s15567), NCOA4 (for in vivo experiment) (Applied Bio systems Life Technologies,s77518), ATG5

(Applied Biosystems Life Technologies, s18159, s18160) and negative control siRNAs (Applied Biosystems Life Technologies, AM4635, AM 4641) were purchased. The *pEGFP-LC3B* plasmid was transfected into BEAS-2B cells using Lipofectamine 2000 (Invitrogen Life Technologies, 11668-027) and stably expressing clones were selected by culturing with G418 (Wako, 070-05183; 1.0 mg/ml) containing medium. Transfections of HBECs and BEAS-2B were performed using the Neon® Transfection System (Invitrogen Life Technologies, MPK5000) using matched optimized transfection kits (Invitrogen Life Technologies, MPK10096).

**Preparation of cigarette smoke extract (CSE)**. Cigarette smoke extract (CSE) was prepared as previously described with minor modifications[44,46]. About 30–50 ml of cigarette smoke were drawn into the syringe and bubbled into sterile PBS in 15-ml BD falcon tubes. We used one cigarette for the preparation of 10 ml of solution. To remove insoluble particles, CSE solution was filtered (0.22 µm; Merck Millipore, SLGS033SS) and was designated as a 100% CSE solution.

**Perls' DAB staining and non-hem iron measurements**. 3,3'-Diaminobenzidine (DAB)-enhanced Perls' staining was used to detect iron accumulation in paraffin embedded lung sections according to the manufacture's instructions. Briefly, sections of lung tissue were washed with PBS and incubated in fleshly prepared Perls' solution (5% potassium ferrocyanide (Sigma-Aldrich)/10% hydrochloric acid) for 1 h, followed by a 15 min incubation in DAB. The labile iron pool of BEAS-2B cells was measured using Calcein AM method as follows. Briefly, $10^6$ BEAS-2B cells or HBEC were recovered using EDTA-free trypsin, counted and incubated in serum-free DMEM with 0.2 µM Calcein-AM at 37 ℃ for 7 min. Cells were subsequently washed and resuspended in Hanks' Balanced Salts solution before the incubation with trypan blue (25 µg). Baseline fluorescence signal was measured at an excitation of 488 nm and emission of 517 nm. The increase in fluorescence upon 100 µM 2,2-Bipyridyl (Sigma-Aldrich, D216305) addition was recorded for each sample[41]. Fe concentration in mouse lung homogenate is measured using inductively coupled plasma mass spectrometry (Agilent technology, 8900 ICP-QQQ) as previously described[27]. Each mouse lung was normalized by protein concentration measured by BCA assay. Conditions for the ICP-QQQ were as follows: RF power was 1550 W. Sampling depth was 8.0 mm. Nebulizer flow rate gas was 1.05 l/min. Cell gas was 7.0 ml/min $H_2$. The isotope which measured was $m/z$ 56. 0.05 mg/l of Co was added to each sample and was used as internal control.

Laser ablation of ICP-MS was analyzed to localize Fe in paraffin embedded mouse lung tissue using a solid state laser ablation system (Electro Scientific Industries, NWR 213). Conditions for the laser were as follows: 10 Hz repetition rate. Spot size was 25 µm. The scan speed was 25 µm/s and 0.8 l/min helium gas flow. Heat maps were generated using imaging software (iQuant2).

Total and ferrous iron in lung homogenates were analyzed by Iron assay kit (Abcam, 83366) according to manufacturer's protocols.

**Measurement of cell death and cell viability**. Cell death was assessed by cytotoxicity analyzed by measuring the release of LDH into media according to the manufacture's protocol (Roche, 4744926), and cell viability determined by MTT assay (Roche, 11465007001) in vitro. TUNEL assay was performed using a DeadEnd fluometric TUNEL system (Promega, G3250) according to manufacturer's instructions. The TUNEL positive cells in lung were detected using fluorescence microscopy (Nikon, Tokyo Japan). The average number of dead cells was assessed by manual counting of TUNEL$^+$ cells in each high power field (×200).

**Measurement of lipid peroxidation in vitro**. HBECs, at a density of $1 \times 10^4$ per well, were seeded in a 96-well microplate. Lipid peroxidation was measured using a C11-BODIPY 581/591 probe (Invitrogen, C10445) as described previously[25]. Briefly, cells were incubated for 30 min with C11-BODIPY 581/591 (1 µM) in growth medium. Fluorescence of C-11 BODIPY was measured by simultaneous acquisition of the green (484/510 nm) and red signals (581/610 nm), providing a ratiometric indication of lipid peroxidation.

**Animal models**. In each experiment, age and sex are matched in each group. 6–8-week-old mice were used in all experiments. C57BL/6 mice were obtained from CLEA Japan, Inc for ICP-MS experiment. Heterozygous GPx4-deficient mice (*GPx4+/−*) and GPx4 Transgenic mice (*TG (loxP-GPx4):GPx4+/+*) on mixed back ground (TT2, ICR, BDF1 strains) were provided by Prof. Hirotaka Imai, Kitasato university[9]. We used *GPx4+/+* on the same background as controls. *GPx4+/+* mice were also used for iron accumulation and NCOA4 expression experiment (Fig. 3a, c, d). All mice were bred in the animal facility at the Jikei University school of medicine. All experimental protocols used in this study were approved by the Jikei University school of medicine Animal Care Committee.

**Cigarette smoke exposure**. Six to 8-week-old mice were exposed using a whole body exposure system (SCIREQ"InExpose") within a barrier facility. Mice were

exposed at a TSP (total suspended particles) of 200 mg/m$^3$ using research cigarette (University of Kentucky 3R4F research cigarettes) for 5 days a week for 1 or 6 months. Age-matched, air-expose mice served as non-smoking controls.

**Morphometric analysis**. Airway morphometry was performed basically as previously described[14]. The left main bronchus was ligated, and the right lung was inflated at a constant pressure of 20 cm for 1 min before fixation in 10% formalin for 24 h. Each of four lobes of the right lung were bisected and paraffin embedded. Histological sections were stained by hematoxylin and eosin, to assess alveolar enlargement. Twenty-one random fields were evaluated by a blinded investigator using digital imaging software (Image J), and alveolar size was estimated by means of the mean linear intercept (MLI) method as previously described[45]. Airway wall fibrosis was assessed by the presence of thick collagen bundles stained by the Pico-sirius red, which expresses wall thickness as a function of area of the airway wall/basement membrane length determined using image analysis software (Image J, v1.36b). Eight to 15 random airways/each group and almost same size (381 ± 64 µm) airways were analyzed by a blinded investigator (Fig. 6).

**In vivo knockdown of NCOA4 in lung using RNAi**. RNAi-mediated NCOA4 knockdown in mouse lung was performed as previously reported[48]. Briefly, individual mice were administered 15 µg each of RNAi agent with In vivo-jet PEI™ (Polyplus Transfection Inc, New York, NY) (resulting in a calculated 1:6 charge ratio of nucleic acid backbone phosphates to cationic lipid nitrogen atoms) in a volume of 25 µl on day 1 using an endotracheally insterted MicroSprayer™ aero-soliser (IA-1C; Penn-Century) and a high-pressure syringe (FMJ-250; Penn-Century, Philadelphia, PA). Each mouse was exposed to cigarette smoke or room air on day 1 to 7 and were sacrificed on day 7.

**Western blotting**. Mouse or human lung tissues were homogenized in ice-cold protein extraction reagent (Thermo Fisher Scientific, 78510) with a complete protease inhibitor cocktail (Roche, 589297001) and phosphatase inhibitor (Roche, 4906845001). Total protein was quantified by BCA protein assay (Thermo Fisher Scientific, 23225). HBECs or BEAS-2B cells grown on 6-well culture plates were lysed in RIPA buffer (Thermo Fisher Scientific, 89900) containing a protease inhibitor cocktail (Roshce, 11697498001) and 1 mM sodium orthovanadate (Wako, 13721-39-6).

Western blotting was performed as previously described[46]. Equal amounts of total protein were resolved by 7.5–10% SDS/PAGE in each experiment. Next, proteins were transferred to polyvinylidene difluoride (PVDF) membrane (Millipore, ISEQ00010), and were incubated with specific primary antibody for 1 h at 37 ℃ or 24 h at 4 ℃. The membrane was incubated with Anti-mouse IgG, HRP-linked secondary antibody, 7076), Anti-rabbit IgG, HRP-linked secondary antibody (Cell Signaling Technology, 7074) followed by chemiluminescence detection (Thermo scientific, 34080, and BIO-RAD, 1705061) with the ChemiDocTM Touch Imaging System (BIO-RAD, California, USA).

**Enzyme-linked immunosorbent assay (ELISA)**. The concentration of IL-1α (R&D, #DY400), IL-33 (R&D, #DY3626), TNF-α (R&D, #DY410) in lung tissue homogenate and HMGB1 (SHINO-TEST, #326054329) in BAL fluid were measured using an enzyme-linked immunosorbent assay (ELISA) kit according to the manufacturer's instructions.

**Immunohistochemistry and Immunofluorescence staining**. Immunohistochemical evaluation of mouse and human lung tissue was performed, as previously described, with a minor modification on paraffin-embedded lung tissues[14].

Immunofluorescence staining was also performed as previouesly described[44].

BEAS-2B cells expressing EGFP-LC3B were grown on 8-well culture slides and were treated with 5% CSE and bafilomycin A1 for 6 h. BEAS-2B cells were then fixed with 4% paraformaldehyde for 15 min followed by permeabilization with 0.03% Triton X-100 (Wako, 16024751) for 60 min. After blocking with 0.1% BSA (Sigma-Aldrich, A2153) for 60 min, the primary and secondary antibodies were applied according to the manufacture's instructions. Confocal lazer scanning microscopy analysis of BEAS-2B cells was performed using rabbit anti-Ferritin Heavy chain, and evaluated by fluorescence microscopy (Carl Zeiss LSM880, Tokyo, Japan).

Fluorescence microscopy analysis of rabbit anti-cleaved caspase-3 and Hoechst 33258 staining were performed in lung sections and evaluated by fluorescence microscopy (Olympus, Tokyo, Japan and Keyence, BZ-X700).

**Sample preparation for LC-MS**. For lipid extraction, PC16:0/16:0-d75 (5 ng) were added to each sample before extraction as an internal standard. Mouse or human lung tissues (0.02 g) were homogenized in distilled water and lipid were then extracted in organic layer by adding a solvent mixture[8].

**LC-ESI-MS/MS system**. The LC-ESI-MS/MS was performed using QTRAP 4500 quadrupole linear ion trap hybrid mass spectrometer (AB Sciex, Concord, ON, Canada) with a Nexera XR high-performance liquid chromatography (Shimadzu

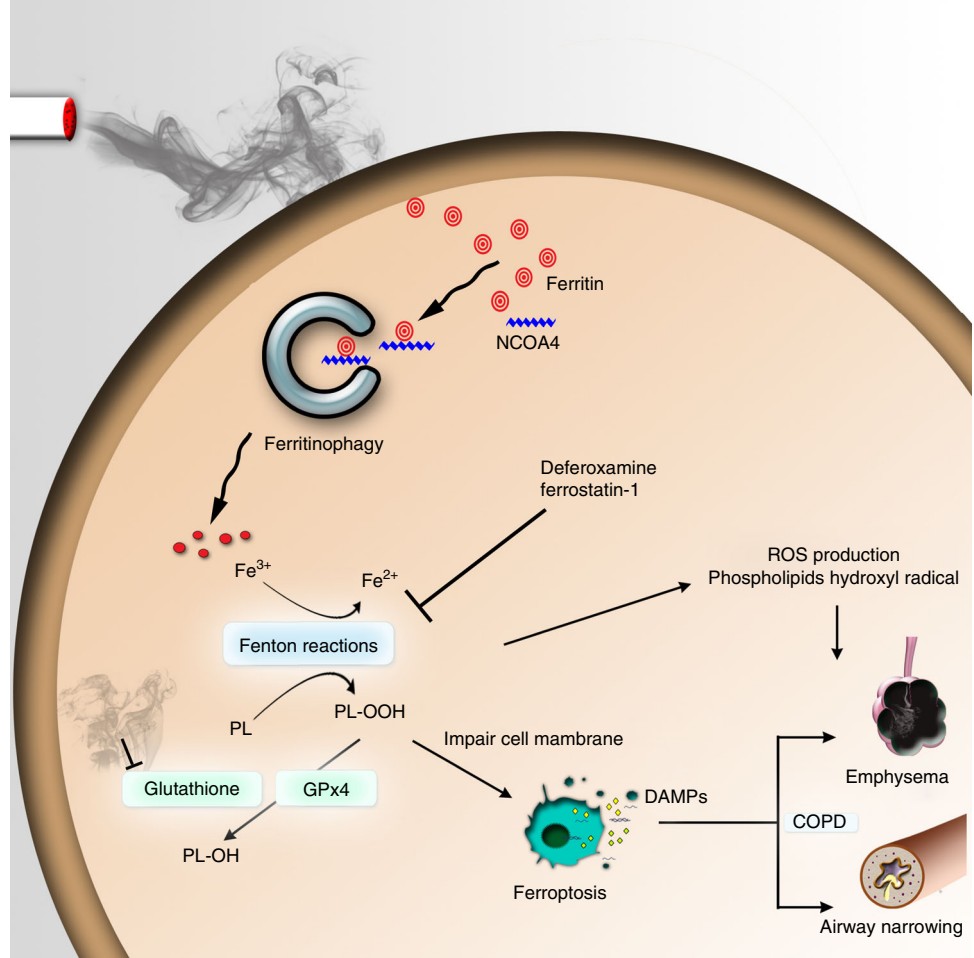

**Fig. 6** Schematic representation of smoking induced ferritinophagy and ferroptosis. Cigarette smoke increases intracellular ferritin and decreases intracellular glutathione transiently. Ferritin is delivered to autophagosomes by NCOA4 and is degraded to non-heme iron via ferritinophagy. The excess of free iron leads to generation of reactive oxygen species and lipid peroxidation through Fenton reactions, which can be regulated by deferoxamine, ferrostatin-1, and GPx4. Lipid peroxidation leads to cell membrane disruption, including in mitochondoria, resulting in ferroptosis. DAMPs and ROS released from ferroptotic cells are implicated in COPD pathogenesis

Co., Kyoto, Japan). The sample was analyzed using the XBridge BEH C18 column (Waters). After the sample was injected to the autosampler, the phospholipid fractions were divided into a step gradient with mobile phase A (acetonitrile/methanol/water = 2:2:1 v/v/v containing 0.1% formic acid and 0.028% ammonia): mobile phase B (isopropanol containing 0.1% formic acid and 0.028% ammonia) ratios of 100:0 (0–5 min), 50:50 (5–25 min), 50:50 (25–59 min), 100:0 (59–60 min), and 100:0 (60–75 min) at a flow rate of 70 µl/min and a column temperature of 30 °C. The multiple reaction monitoring was performed to detect specific oxidized phospholipids. MS/MS analysis was performed in negative ion mode with the following settings, ion spray voltage, −4500V; curtain gas ($N_2$), 30 arbitrary units; collision gas ($N_2$), 'medium'; declustering potential, −60 to −170 V, collision energy, −40 to −44 eV; temperature, 500 °C.

In order to detect PC, formate adduct ions ($[M + HCOO]^-$) were used as precursor ion, and the fatty acyl chain were used as product ions ($[M−H]^-$). In order to detect PE, deprotonated ions ($[M−H]^-$) were used as precursor ion, and the fatty acyl chain were used as product ions ($[M−H]^-$). In either case, peroxidized fatty acyl chain were detected as water-loss ion ($[M−H−H_2O]^-$).

**Transmission electron microscopy**. Electron microscopy was performed as previously described[44]. Lung tissue from pneumonectomy and lobectomy specimens and HBECs transfected with siRNA and treated with CSE were fixed with 2% glutaraldehyde/0.1 M phosphate buffer (pH 7.4) after 48 h of incubation and dehydrated with a graded series of ethanol. Fixed HBECs were then embedded in epoxy resin. Ultrathin sections were stained with uranyl acetate and lead citrate and observed with the Hitachi H-7500 transmission electron microscope (Hitachi).

**Statistics**. Data are shown as the mean ± SEM taken from at least three independent experiments. Student's *t*-test was used for comparison between two different groups, while analysis of variance was used for multiple data sets. Turkey's test were used for parametric and nonparametric data, respectively, to find where the difference lay. Outliers were determined by Grubb's test and removed (GraphPad Software). Significance was defined as $p < 0.05$. Statistical software used Prism v.6 (GraphPad Software, Inc., San Diego,CA)

**Study approval**. Informed consent was obtained from all surgical participants as part of an approved ongoing research protocol by the ethical committee of Jikei University School of Medicine in full accordance with the declaration of Helsinki principles. Animal experiments were approved by the committee on Animal Care at th Jikei University and performed in accordance with institutional guidelines.

**Reporting summary**. Further information on research design is available in the Nature Research Reporting Summary linked to this article.

## Data availability
All relevant data supporting the key findings of this study are available within the article and its Supplementary Information files or from the corresponding author upon reasonable request. Source data for human and mouse MS experiments and uncropped images of blots are provided as a Supplementary Source Data file.

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

## Acknowledgements

The authors thank Agilent Technologies Japan, Ltd for technical assistance with ICP-MS. We also thank Stephanie Cambier for comments on the manuscript. This work was supported in part by Grants-in-Aid from Scientific Research (C) (17K09673, 26460075) from JSPS KAKENHI, Scientific Research on Innovative Areas (16H01367 and 17H05513) from a MEXT (Ministry of Education, Culture, Sports, Science and Technology), Japan, GlaxoSmithKline research grant (2016), and AMED under Grant Number, JP18gm0910013.

## Author contributions

M.Yo. designed and performed the experiments and revised the manuscript. H.H., K.T., Y.H., A.I., Nay.S., T.K., Nah.S., Y.K., Ke.K., S.I. helped the cell culture and bleeding animals, and participated in the discussion. J.A. and Ka.K. supervised the research and revised the manuscript. H.U., H.W., T.N., Y.K., K.N.participated in the discussion. Sho.M., H.A., M.Ya., M.O., T.M. provided patient samples. T.I provided helpful comment about MS analysis. T.S. performed LC-MS analysis. H.I. provided *GPx4*+/−, *GPx4*TG mice and helpful comments. Shu.M. designed the project and wrote the manuscript.

## Additional information

**Competing interests:** The authors declare no competing interests.

