## [Peer Review File · Nature Communications]

Reviewer #1 (Remarks to the Author):

Yoshida et al provide a fascinating manuscript on the role of ferroptosis and
NCOA4-mediated ferritinophagy in the effects of cigarette smoke and development
of COPD. They analyze cell culture models, animals models and patient samples,
and come to a consistent conclusion that CSE increased iron abundance by
upregulating NCOA4-mediated ferritinophagy, leading to ferroptosis. Overall, the
study is well performed with suitable controls, and well described. Some of the
effects are modest in cell culture, but statistically significant and will be of
interest to readers. My only concern is that many of the figures are very small
and low quality, so it is hard to draw firm conclusions from them. I suggest that
the authors need to redo the figures to make them large enough and high enough
resolution to be clearly seen by readers.

*We thank the reviewer for a thorough review with constructive comments. According to*
*reviewer's comment, we have replaced low quality figures to high quality(resolution) ones.*

Reviewer #2 (Remarks to the Author):

In this manuscript, Yoshida et al studied the potential role of ferroptosis in
a cigarette smoke (CS)-induced chronic obstructive pulmonary disease (COPD). The
authors showed in a cellular model that CS can induce ferroptotic cell death, which
is positively regulated by NCOA4-mediated autophagy targeting the iron-storage
protein ferritin (ferritinophagy). The authors further showed that ferroptosis
plays a crucial role in COPD pathogenesis in a mouse model. They also found that
increased lipid peroxidation and NCOA4 expression levels are associated with COPD.

Overall, the authors provided convincing evidence to demonstrate the pathological
relevance of ferroptosis to human COPD. However, the in vivo (animal model)
evidence for the role of NCOA4-mediated ferritinophagy in CS-induced COPD is
lacking. If the authors can perform in vivo experiments to demonstrate that
blockage of NCOA4-mediated ferritinophagy is indeed able to decrease CS
induced-iron deposition, cell death and other pathological phenotypes, their

central hypothesis will be more convincingly supported, and the significance of
this study will be greatly improved.

*We really appreciate your thorough review and supportive comments. We agree with your*
*comments. NCOA4 knockout mouse has been reported to show impaired systemic iron*
*homeostasis, including iron accumulation in the liver and spleen. (Bellrlli R et al. Cell Reports*
*14,411-421 Jan 2016). Therefore, several organ damages may affect NCOA4 KO mice viability*
*and NCOA4 KO mouse may not appropriate model to demonstrate the preventive role in*
*CS-induced COPD phenotype development. If we can prepare lung epithelial cells specific*
*NCOA4 KO mice (NKx2.1Cre NCOA4 Flox mice), that will be a great model to confirm our*
*experimental results and physiological involvement of NCOA4 in COPD pathogenesis.*
*However, it will take more than 6 months to generate this phenotype and additional 6 months*
*are needed to perform CS exposure experiments. So, we would like to work on this task as the*
*next project.*

Reviewer #3 (Remarks to the Author):

Review of NCOMMS-18-04233: Involvement of cigarette smoke-induced ferroptosis via
NCOA4-mediated ferritinophagy in COPD pathogenesis

Summary

The authors of this study propose that ferroptosis, a newly defined mode of lipid
and iron mediated programmed cell death contributes to the pathogenesis of COPD.
In this study the authors demonstrate that cigarette smoke extract, an in vitro
model for COPD activates ferroptosis in human bronchial epithelial cells. The
authors demonstrate that such an activation of ferroptosis in epithelial cells
could be prevented by the addition of the ferroptosis inhibitor ferrostatin-1 and
the extracellular iron chelator deferoxamine. The authors speculate that one
mechanism for smoke-induced ferroptosis in bronchial epithelial cells may involve
the ferritinophagy adaptor protein NCOA4. The authors demonstrate that NCOA4 is
activated by smoke in cultured bronchial epithelial cells as well as in whole lung
mouse tissues. They hypothesize that such an activation of ferritinophagy in
epithelial cells leads to the increased breakdown of ferritin and subsequent
release of iron in response to smoke.

Using an in vivo model of smoke-induced COPD, the authors demonstrate that mice
with a heterozygous deletion of a negative regulator of ferroptosis, GPX-4
exacerbates smoke induced injury and inflammation whereas mice with transgenic
activation of GPX-4 are protected from smoke induced COPD. Finally, they end the
paper by examining the association of NCOA4 with clinical lung function decline
in patients with COPD as well as demonstrating the association between an increase
in lipid peroxidation products and presence of disease in these patients.

*We thank the reviewer for a thorough review.*

General Comments

This is an interesting study with translational relevance demonstrating clinical
prognostic or therapeutic applicability for human COPD. It is a well-written study
by distinguished well-published investigators in the COPD/Lung Disease field,
albeit the role of iron metabolism in COPD pathogenesis is a new field for this
principle investigator. The murine models used in this study are experimentally
sound and are adequately comprehensive. The novel finding that ferroptosis and
ferritinophagy are activated in bronchial epithelial cells in vitro and in vivo
in response to smoke are strengths to this paper, as well as the use of the Gpx4
heterozygous and Gpx4 transgenic mice. These results are in line and build on
previous observations highlighting a role for abnormal iron loading in the
pathogenesis of COPD.

*We really appreciate your thorough review. The comments were constructive and the merit of*
*the work was clearly appreciated and the data was obviously carefully inspected.*

While this is a nice study, the mechanistic link between NCOA4 and ferroptosis
presented by the authors is somewhat flawed. Of main concern is why, when the
authors focus on ferroptosis as the major concept for this paper, they fail to
demonstrate if GPX-4 is regulated by smoke in any of their model systems, in vitro
or in vivo.

*We appreciate this thoughtful comment. As pointed out by reviewer, we observed no significant*
*correlation between COPD severity and GPx4 expression levels in lung homogenates in human*
*lung samples and CSE treatment did not modulate GPx4 expression levels in HBEC, which*
*were added a new Supplementary Figure6. Furthermore, GPx4 expression levels appears to be*
*elevated in smoke exposed mice lung compared to room air control lungs, indicating that GPx4*
*levels may be upregulated in COPD lungs as a part of adaptive response to not only increased*
*lipid peroxidation but also reduced GSH(gluthatione). GPx4 utilizes the reducing equivalents*
*from its substrate GSH and it has been reported that GSH levels were clearly reduced in*
*response to CS (Exp Toxicol Pathol. 2013 Sep ;65(6):711-7 , Environ Toxicol*
*Pharmacol.2017 Sep;54:40-47). Therefore, we speculate that reduced GSH levels can be linked*
*to insufficient GPx4 activity, resulting in excessive lipid peroxidation with concomitantly*
*increased ferroptosis. To clarify this point, we also measured the levels of GSH during our in*
*vitro experimental models. In line with previous findings, we observed clear reduction of GSH*
*in response to 24hr CSE exposure in HBEC (new supplementary Fig10h,i). Furthermore, NAC*
*treatment increased GSH levels, further supporting the notion that not only GPx4 levels but*
*also GSH levels are critical determinant for regulating lipid peroxidation and subsequent*
*ferroptosis (new supplementary Fig10h,i). Based on this finding, we speculate that GPx4*
*modified mouse models may not directly mimicking COPD pathology, but clearly reflecting the*
*pathogenic condition of increased lipid peroxidation in COPD lungs, which can be attributed to*
*reduced GSH of insufficient GPx4 activity. To clarify this point, we added the experimental*
*results of GSH assay in New supplement Figure 10 and add the sentences to discussion section*
*as following.*

In addition, the authors do not demonstrate a direct link between ferroptosis
mediated by GPX4 and NCOA4. Why choose NCOA4 as a mechanism? When the lipid
peroxidation story, a more concrete mechanistic rationale for ferroptosis is not
developed at all? Does GPX-4 expression in the bronchial epithelium correlate with
disease in human COPD patients?

*We appreciate this important suggestion. We think that the direct link in expression levels*
*between GPX4 and NCOA4 may not be necessary for conducting ferroptosis based on the*
*observation of GSH. However clear positive correlation between NOCA4 expression levels in*

*lung homogenate and pulmonary function tests indicated that NCOA4-mediated iron*
*metabolism may have an essential role in COPD pathogenesis via enhancing ferritinophagy.*
*It is likely that enhanced ferritinophagy-mediated release of free iron can be a critical*
*determinant for lipid peroxidation through Fenton type reaction. Therefore, direct association*
*between GPx4 and NCOA4 expression levels may not exist, insufficient GPx4 activity of*
*reduced GSH and increased NCOA4-mediated ferritinophagy may functionally linked in terms*
*of enhancing lipid peroxidation and ferroptosis. Therefore, we selected NCOA4 as a target*
*molecule for regulating ferroptosis during COPD pathogenesis.*

The bigger picture concept that ferroptosis inhibitors may have therapeutic
applicability in COPD is nice, novel and highlighted in this study. However,
ferrostatin-1 is a lipid ROS scavenger, that is not thought to chelate iron (Cell.
2012 May 25; 149(5): 1060-1072) and so it is unclear why the authors did
not focus in on lipid peroxidation pathways but rather focus in on iron (which
is also not that well developed) and the turnover of ferritin as a mechanism for
this study.

*We appreciate this thoughtful comment. We believe free iron release by ferritinophagy is the*
*key step for lipid peroxidation with subsequent ferroptosis. Hence both free iron and lipid*
*peroxidation can be therapeutic target with respect to modulating ferroptosis. However we*
*observed clear inhibitory role of DFO in CSE-induced lipid peroxidation and ferroptosis, DFO*
*may demonstrate toxic effect on in vivo mouse model, hence we selected ferrostatin-1 to*
*elucidate the involvement of ferroptosis in COPD development in mouse models.*

Moreover, while there is clear obvious difference between the response of the
GPX-4 TG and GPX-4 heterozygous mice to smoke, the authors do not address the
important question of whether the bronchial epithelial cell is the main player
in this phenotype. While bronchial epithelial cells in the lung have been shown
to express high levels of

GPX-4 (<https://www.proteinatlas.org/ENSG00000167468-GPX4/tissue>), the

expression of NCOA4 is quite low in these cells

(<https://www.proteinatlas.org/ENSG00000266412-NCOA4/tissue/bronchus>).

Deleting GPX-4 using the loxp system with a bronchial epithelial cell specific

cre would have really addressed the role of GPX-4 in bronchial epithelial cell
biology. I am not sure if NCOA4 has the same important effect in these cells.

*We really appreciate this thoughtful remark. Although we used GPx4 modified mouse models to*
*clearly show the involvement of lipid peroxidation and ferroptosis in COPD development, as*
*described above no clear reduction of GPx4 was detected in COPD lungs and CSE-treated*
*HBEC. Therefore, lung epithelial cell specific GPx4 KO mouse may clarify the epithelial*
*specific role of GPx4 during COPD mouse models, but those models may not add more*
*information regarding physiological involvement of ferroptosis in human COPD pathogenesis.*

Finally, the novel finding that smoke activates this recently defined form of
programmed cell death is not surprising. This study is an additive study to the
ever-growing literature on the role of yet another cell death pathway in
smoke-induced COPD. In the last 15 years there have been many studies showing that
had cigarette smoke induces apoptosis, necroptosis, autophagy-mediated cell death
as well as necrosis in many different lung cell types. While the authors claim
in this study that ferroptosis and not necroptosis is important in their conditions,
they should not be too quick to eliminate necroptosis or all other forms of cell
death and hence confuse the field even more. Indeed, all of the programmed and
non-programmed cell death types maybe observed upon smoke treatment, which will
be dose dependent, time dependent and cell type dependent.

*We appreciate this important suggestion. We totally agree reviewer's comment and to avoid the*
*over statement, we delete the results of necroptosis in revised version of manuscript.*

An example of interest would be the fact that in the past publications written
by this group concentrations of 1% smoke extract have been used, but for this
particular current study 5% CSE is used. I wonder if the authors observe
ferroptosis readouts in their 1% doses too? None the less, programmed cell death
pathways are of interest to lung biologists to really understand how cell
populations respond and protect themselves from stress.

To summarize, this study has good translational relevance and contains some novel
interesting concepts. However, this study has some fatal mechanistic flaws that

preclude my enthusiasm for this manuscript. A more focused study on the lipid
peroxidation angle and ferroptosis with GPX-4 and in turn the association of this
pathway with human COPD would enhance the focus and novelty of this paper. In
addition to the above comments, I have a number of specific major concerns with
this manuscript in its current form.

*We appreciate this thoughtful remark. In terms of CSE concentration, no obvious cytotoxic*
*effect was demonstrated by using 1% CSE but cellular senescence was induced. Does dependent*
*cytotoxic effect was observed during CSE treatment and 5% CSE was sufficient to detect*
*significantly increased cell death, thus we selected 5% CSE in our experimental condition.*
*As described above, we believe the existence of causal link between NCOA4-mediated free-iron*
*release and enhanced lipid peroxidation for ferroptosis. We believe that reduced GSH levels*
*may indirectly indicate insufficient GPx4 activity to prevent lipid peroxidation and ferroptosis*
*in the setting of enhanced NCOA4-mediated free-iron release during COPD pathogenesis.*

Specific Major Comments

1. What is the expression of GPX-4 in human COPD and the association with
FEV1?

*We appreciate this important comment. Significant correlation between COPD severity and*
*GPx4 expression levels was not shown in human lung homogenates. We have added this result*
*in new Supplementary Figure6.*

2. The authors do not comment on whether GPX-4 is modulated by smoke in any
of their models?

*We appreciate this important comment. GPx4 expression in HBEC didn't modified by CSE*
*treatment. We have added this experimental result in new Supplementary Figure6.*

3. Do the GPX-4 heterozygous mice or the GPX-4 transgenic mice show altered
NCOA4 or ferritin expression levels in response to smoke? Do these mice show
altered iron content inside bronchial epithelial cells?

*Neither NCOA4 nor ferritin was altered by GPx-4 heterozygous mice or the GPX-4 transgenic*
*mice, suggesting that GPx4 works downstream of NCOA4-mediated free-iron release and does*
*directly regulate NCOA4 and ferritin expression levels. We have added these results in new*
*Supplementary Figure3c.*

4. Ferritin accumulation at 1-9h in Fig1 but also an increase in NCOA4 at
1-9h -if ferritin accumulating then you would expect Ncoa4 to be
down-regulated-can the authors speculate on this observation?

*Do you mean Figure2a? I don't understand the meaning of this question.*

*In the present study, peak of Ferritin expression is earlier than NCOA4 expression peak. After*
*24hr ,(36hr), NCOA4 was slightly decreased (data not shown).*

5. There are many other proteins that regulate ferritin activation and
repression; namely the two-master iron regulatory proteins Irf1 and Irf2. Can the
authors prove that the increase or loss of ferritin observed in Figures 1 and 2
is due to NOCA4 or does this also involve the IRP system?

*Thank you for your suggestion. We don't exclude the potential involvement of IRP proteins for*
*ferritin regulation in COPD pathogenesis. In our new experimental data showed that IRP-2 was*
*elevated by CSE exposure in HBEC (revised supplementary Fig10. When we restrained*
*NCOA4 using siRNA, IRP2 expression was marked upregulated in HBEC (revised*
*supplementary Fig10g), suggesting that NCOA4 may be interacted with IRP2 to maintain iron*
*homeostasis in HBEC. However, we appreciate if we can focus on NCOA4 in the present*
*study because some more intracellular iron regulators may interacted with each other including*
*transferrin and ferroportin, which were upregulated in response to CSE exposure in a time*
*dependent manner(revised supplement Fig.10). We believe that NCOA4-mediated*
*ferritinophagy is a novel mechanism of iron regulation in COPD pathogenesis.*

6. Concentrations of CSE used in this study. Cell studies involving
cigarette smoke extract are completely dependent on the concentration and
preparatory methods used to make the extract. In this study, the authors use 5%
CSE throughout preparing this extract by bubbling 1 cigarette per 10ml. This
preparatory method is comparable to concentrations used by other investigators
in this field. However, in prior publications by this group, only 1% CSE
concentrations have been used. For example, in the authors 2012 oncoimmunology,
2013 Am J Physiol Lung Cell Mol Physiol and 2015 autophagy papers the authors only
use 1% CSE? Can the authors comment on why they change concentrations now? Also,
in Figure 1. The authors use a 24hour exposures to show lipid peroxidation and
protective effects of DFO but to really get an idea of whether we are just seeing
concentration dependent cell mechanisms it is important the authors show the dose
response of CSE as well as the time responses.

*Thank you for providing these insights. Our previous study demonstrated that 1%CSE treatment*
*lead to cellular senescence in HBEC. In terms of HBEC ,24hr exposure of 1%CSE induces*
*senescence, and 5%CSE induces cell death in light of our past experience.. As showing below,*
*cytotoxicity was increased by CSE in a dose- dependent manner ,but 10% was too toxic to*
*survive for HBEC.*

Figure 2C- what was the time of CSE exposure here?

*I have addressed "24hr" in Figure legend.*

7. DFO is an extracellular iron chelator-it does not traverse cell membranes
and so how do the authors speculate it prevents ferroptosis if it cannot chelate
the iron on the inside? Specifically, DFO is an extracellular iron chelator

(siderophore) that has trouble traversing cellular membranes. DFO has a low lipid
solubility (K_{part} of 0.001) and a molecular weight of more than 657 preventing
its uptake into cells (Ann N Y Acad Sci. 2005;1054:155–68 and Br J Haematol. 1994
Apr;86(4):851–7). However, it is positively charged and so some cells are able
to take up DFO—however not usually faster than 1hour. In this study the authors
use 100uM DFO pre-treated for one hour before the addition of CSE (5%) for 24hours.
If we presume that DFO is not taken up into the cell that fast to prevent any
intracellular changes occurring in the labile iron pool of the cell, to prevent
ferroptosis, then how do the authors suggest DFO is preventing CSE-induced cell
death? It may be more likely that the
DFO pretreatment is chelating any of the iron present in the smoke extract and
thereby preventing iron or CSE particles from entering the cell.

*Can the authors show that within 1hour pretreatment with DFO that the labile
iron pool is altered inside the cell—irrespective of the CSE? * Can the authors
measure the amount of iron in their CSE extract? *Is the DFO limiting the
activation of NCOA4, or altering the expression of GPX4?

In the final figure of this paper, the cartoon depicts that DFO is inside the cell
and modulating ferroptosis— the authors should be aware that putting DFO inside
the cell is not correct and quite misleading to the reader unless the authors can
prove otherwise.

It may be more likely in this case that DFO is mediating smoke induced cell death
by (as mentioned above) altering the metal concentration of smoke extract or by
activating protective HIF-1 alpha signaling cascades, or by altering Irf1 or Irf2.
If the authors could show DFO was inhibit lipid peroxidation this would certainly
provide more rational to use this molecule to study ferroptosis.

*You have raised an important point. We think that it is controversial if DFO chelates only*
*extracellular iron. Previous paper indicates that DFO enters cells by fluid-phase endocytosis*
*and exerts its effects by chelating redox-active iron present in the endosomal/lysosomal*
*compartment. (Doulias PT et al. Free Radic Biol Med 2003 Oct 1;35(7);719-28). Furthermore,*
*in our revisions, 1hour pretreatment of DFO significantly reduced labile iron pool in HBEC,*
*whether CSE is treated or not, suggesting DFO enter HBEC and chelates labile iron in our in*
*vitro experimental condition.*

*We have added this result in supplementary Figure 10a.*

“the amount of iron in their CSE extract”

*We have not analyzed the amount of iron in their CSE extract in the present study. Previous*
*report showed the iron contents were small but present in gas and tar extracts of cigarette*

*(0.04±0.02 and 0.13±0.04 nmol of iron/cigarette smoke).(Am J Respir Crit Care Med*

*Vol151.pp431-435,1995). Actually the amount of iron in the medium with CSE is considered to*
*be unstable ,because dead cells may release ferritin or iron in the medium.*

“Is the DFO limiting the activation of NCOA4, or altering the expression of
GPX4?”

*As showing below, DFO did not altered GPx4 and NCOA4 expression.*

8. Does liproxstatin-1 (Lip-1) also work in the authors hands at preventing
smoke-induced death?

*We have performed Liproxstatin-1 treatment experiment in our revision. Lip-1 significantly*
*prevented smoke-induced cell death. We have added this result in supplementary Figure 10b,c.*

9. Ferrostatin-1 may display prophylactic protective responses under
conditions of in vitro smoke extract treatment. Many antioxidant-like molecules
that have show little promise in the treatment of human COPD as the smoke
antioxidant effects are long gone- unless the authors are proposing to administer
this drug in combination with smoke?

*Ferrostatin-1 may be a new therapeutic candidate for COPD. However, to speculate the*
*clinical usefulness, in vivo experiment for Fer-1 treatment should be performed.*

10. The labile iron measurements in this paper are nice, however these
experiments only represent the free iron pool and do not give us any indication

about the total iron burden of the cell, including total iron and non-heme iron
levels. Other readouts for the iron status of the cell would be useful including
transferrin receptor activity, Irf1/2, ferroportin etc.

*We analyzed transferrin receptor , IRP-1/2,and ferroportin expression level in response to CSE*
*exposure in HBEC. CSE led to a time-dependent increase in IRP-2, transferrin ,and ferroportin*
*expression. These data may partly explain the total iron uptake . We included this results in*
*supplementary Fig.10d. and the result section.*

11. Cell lines used. In Figure 1 the authors carry out all experiments in HBEC
cells but then switch to Beas2B cells in Figures 2—either pick one or the other—and
make sure all experiments are performed in the one cell line. In an ideal situation
using mouse tracheal epithelial cells grown at an air liquid interface would be
a more appropriate model—but HBEC is OK.

*We reperformed some crucial experiments in Figure 2 by using HBEC ,and compared with the*
*results of experiments using BEAS2B. NCOA4 timecourse ,Lipid peroxidation assay, and*
*cell-death experiments using HBEC showed similar results in BEAS2B cells. We included these*
*results using HBEC in supplementary Figure 10.*

12. Previous studies have demonstrated that excessive iron inside alveolar
macrophages (Philippot et al PLoS One. 2014 May 1;9(5):e96285) associates with
disease severity in COPD— with macrophages being the dominant iron stained cell.
While others have shown iron loading inside epithelial cells, I am unaware of
anyone associated iron staining in bronchial epithelial cells and disease severity
in COPD—can the authors demonstrate that patients with COPD have higher iron
staining in bronchial epithelial cells compared to healthy smokers and controls?

*We have added new panels of 4-HNE and Perl's staining by using healthy smoker's lung in Fig*
*5a. 4HNE was stained in healthy smoker ,stronger than never smoker but weaker than COPD.*

*Perl's staining was not detected in health smoker's lung.*

Additionally, in Figure 5a the authors conclude that the amount of iron inside
bronchial epithelial cells in the lung tissue of COPD patients associates with
4HNE staining– however this is not clear in Figure 5a? A better staining approach
to show this apparent concomitant or co-staining patterns
staining of 4HNE and Perl's stain in bronchial epithelia cells is needed.
*Due to the technical issue, immunofluorescence of 4HNE and Perl's staining did not work well*
*until now. We have performed experiments by the modality of serial section staining. To clearly*
*demonstrate stained cells, We have replaced corresponding pictures in Figure 5.*

13. NCOA4 staining in Figure 5b– this is in whole lung homogenates? Can the
authors show NCOA4 staining in lung sections from smokers in addition to the
controls and COPD patients?

*Yes. Fig5b is whole lung homogenates. We added NCOA4 staining from healthy smoker's lung*
*in Fig5d.*

14. Initially when reading, I was unsure of the relevance of showing
accumulation of small sized mitochondria with increased membrane density
was obvious in airway epithelial cells in COPD lung, but was barely detected in
airway epithelial cells in non-smoker lungs (Fig. 5g)–in the discussion,
this rationale is give, as this has been demonstrated by others in the context
of ferroptosis–may be make this a bit clearer in the text of the results section
before describing this phenomenon

*Thank you for your suggestion. We have incorporated “Morphological features of ferroptosis in*
*transmission electron microscopy (TEM) have been represented by dense and smaller*
*mitochondria with increased membrane density and vestigial cristae^{6 10} in the*
*results(Line131-133). Further, we added “In good agreement with recent reports associated*
*with ferroptosis” before this sentence(Line264).*

15. For the GPX-4 transgenic mice– the original paper referenced that made
these mice states the mice were generated on a mixed background? Yet the authors
state here the mice are on a C57/BL6 background? Please clarify– if these mice

were on a mixed background then the appropriate controls must be used in addition
to the C57/BL6 controls used here

*We thank you for pointing out. It's my mistake. I confirmed the background of these mice to*
*Dr.Imai who generated GPx4 knock out and transgenic mice. These mice were Mixed back*
*ground . They have been interbred each other more than 10 times. Also ,we used GPx4+/+ mice*
*as negative control, which is on same back ground as GPx4+/- and GPx4 transgenic mice . I*
*corrected this point on discussion of the manuscript.*

16. I am not sure how adding the RIPK3 data is helpful here unless you plan
to also show that apoptosis deficient mice or autophagy deficient are also not
protected from smoke?

*Thank you for providing these insights. We agree with you and excluded supplement Fig 6 (WT*
*vs RIP3KO mice in response to CS 6 months).*

Minor comments

1. Please make sure that throughout when referencing prior papers related
to smoke what cell types the authors used (e.g. line 144–145). There is a clear
confusion in the field about the role of these various different cell death
pathways, e.g. Autophagy being activated in epithelial cells but decreased in
alveolar macrophages after smoke and this is partly due to the lack of clarity
in the cell models used i.e. epithelial cells versus endothelial cells versus and
immune cell- please state the cell type whenever mentioning prior findings by
others

*You have raised interesting point; Various different cell death pathways may be detected in*
*Various different cell types. However,we believe that aberrant autophagy and cell death in lung*
*epithelial cells ,a initial response during cigarette smoke exposure, is a crucial step in COPD*
*pathogenesis. Hence, we focused on lung epithelial cells. Referencing prior papers (Line*
*144-145) used A549 (lung alveolar cell line). We revised this point.*

2. Figure 3 A and 5A– Perl's stain is spelt Perl's; not
Perl's– in Fig 3a/5a there needs to be some counter stain for cytosol to
see the cell structures better. Enhanced zoomed in regions should also be shown.
The resolution of the images is really bad.

*“Perl's DAB staining” is to be used consistently. We have added counter staining with fast red*
*on Figure 3a. Figure 5a are serial section staining and there are counter staining with 4-HNE*
*staining . So, we have changed each images more zoomed easy to detect cell types.*

3. Resolution of all images in Figure 5 is bad– I cannot see the localization
of NCA04 in COPD tissue– please also show healthy smoker tissues in addition to
the non-smoker controls throughout.

*We have replaced all images to high resolution version. We have added new panels of 4-HNE,*
*Perl's DAB ,and NCOA4 staining by using healthy smoker's lung in Fig 5.*

4. A comment should be added in the discussion that different doses of
cigarette smoke extract may elicit different types of cell death at different time
points and so it is not as clear cut to say one type of death is predominant–this
also maybe cell type specific

*Thank you for your suggestion. We added your sentence in the discussion.*

5. Figure 4h–quantification of TEM needed here

*We performed quantification of abnormal mitochondria and added the graph on Figure4h.*

6. For the ICP–MS Fe work–how were the measurements normalized–protein?
Please outline in the methods and in the figure legends –or make sure the reported
numbers are normalized quantities

*We have added these sentence as below(red) in the methods.*

*“Fe concentration in mouse lung homogenate is measured using inductively coupled plasma*
*mass spectrometry (Agilent technology, 8900 ICP-QQQ) as previously described(27).*

*Each mouse lung were normalized as protein concentration measured by BCA assay.*

*Conditions for the ICP-QQQ were as follows: RF power was 1550W.*

*Sampling depth was 8.0mm. Nebulizer flow rate gas was 1.05L/min. Cell gas was 7.0mL/min*
*H₂.*

*The isotope which measured was m/z 56. 0.05 mg/L of Co was added to each sample and was*
*used as internal control.”*

7. It is not clear if the mice used in these experiments were sex matched
and the distribution of males to females –more details are required in the methods
section.

*We have corrected “animal models” in the methods.*

*We look forward to hearing from you regarding our submission. We would be glad to respond to*
*any further questions and comments that you may have.*

*Dear reviewers,*

We thank referees for careful reading our manuscript and for giving insightful comments.

We have incorporated referees and editors comments ,and revised our manuscript as
possible.

*In the revised manuscript, first revised place is highlighted yellow, and 2nd revised
place is highlighted green.

Our responses to the referee's comments are as follows:

Reviewers' comments:

Reviewer #2 (Remarks to the Author):

The authors have sufficiently addressed the concern of this reviewer, except that in the
revision they still have not established the in vivo function of NCOA4. On the other
hand, this reviewer does understand their reasoning of lack of the in vivo data (that to
demonstrate the in vivo relevance of NCOA4 is time-consuming and might be beyond
the scope of this study). Therefore, this reviewer will accept the lack of this important
data but request the authors to unambiguously state in the paper that while all
observations are consistent with the proposed role of NCOA4, the definite in vivo
causative evidence is lacking.

*Thank you for your suggestion. Associate Editor also concerned about lacking of*
*NCOA4 experiment in vivo, hence we have attempted to performe in vivo experiment by*
*using NCOA4 knockdown models.*

*WT(GPx4+/+) mice were exposed to whole body mainstream CS for 7days. Control*
*siRNA or NCOA4 siRNA was injected intra-tracheally by using in vivo-jetPEI® on day1.*

*IHC staining demonstrated that NCOA4 siRNA injection efficiently diminished NCOA4*
*expression levels in bronchial epithelial cells. We evaluated ferritin and 4HNE*
*expression in bronchial epithelial cells, which may reflect ferritinophagy and lipid*
*peroxidation. We used consecutive sections of mouse lung tissues for detecting ferritin*
*and for 4-HNE. Ferritin expression was enhanced in NCOA4siRNA treated BEC*
*compared to control siRNA treated BEC, suggesting NCOA4 siRNA efficiently blocked*
*ferritin degradation of ferritinophagy in BEC. 4HNE expression was significantly*
*decreased in NCOA4siRNA treated BEC compared to control siRNA treated BEC,*
*suggesting blocking ferritin degradation by NCOA4siRNA inhibited lipid peroxidation ,*
*which can be attributed to reduced Fenton type reaction. Next, dead cells count by*
*means of TUNEL assay in mouse lung were performed. TUNEL positive cells were*
*significantly increased by CS exposure ,which were significantly decreased by*
*NCOA4siRNA treatment.*

*We understand the potential limitations of our in vivo experimental models using short*
*time period CS exposure, because this siRNA delivery system is also able to knockdown*
*for short term in BEC. Thus, we could only evaluated cell death but could not assess the*
*alteration of COPD phenotype by NCOA4siRNA injection model. However, we believe*
*that these results strongly support our hypothesis, “NCOA4-mediated ferritinophagy*
*induces ferroptosis via Fenton type reaction especially in the setting of reduced GPX4*
*in COPD pathogenesis” We added these figures in Supplementary Figure 6 .*

Reviewer #3 (Remarks to the Author):

I thank the authors for their comprehensive response to my concerns. I have carefully
read each of their responses. Of note, the liproxstatin-1 (Lip-1) data is a nice addition
and supports the lipid peroxidation angle as a nice mechanism to this story. However, I
still feel that there are a number of fatal mechanistic concepts in this paper that, in my
opinion, do not fit the caliber of this particular journal, as outlined below.

Major

1. The focus of this paper is on ferroptosis, yet the authors reveal in the revised
version (as I had requested) that there is no significant correlation between
COPD severity and GPx4 expression levels in lung homogenates in human lung

samples and CSE treatment did not modulate GPx4 expression levels in
HBECs; The authors actually see an increase in GPX4 in their murine models,
an observation that does not fit the current dogma that loss of GPX-4 expression results
in the activation of ferroptosis. The authors suggest that this may be due to loss in the
activity of the protein, but they do not provide any concrete evidence for this. The
phenotype in their GPX-4 mice may therefore not be anything to do with ferroptosis and
so how can the authors make this paper about ferroptosis when they do not see any
evidence for it in their human samples or in their in vivo or in vitro models?

*We have revised our previous comment by adding our new experimental results*
*regarding this point. In the initial manuscript, we have measured GPx4 expression*
*levels by using lung homogenates, which contains a wide variety of cell types (including*
*inflammatory cells, which may express high levels of GPx4 based on IHC evaluation)*
*other than epithelial cells. Furthermore, COPD lung samples used for lung*
*homogenates were mild-moderate disease severity. We speculated that methodological*
*and sample errors affected on those negative results. Hence, to clarify the pathological*
*link between GPx4 expression levels in HBEC and COPD progression, we have*
*obtained more severe (advanced) COPD lung samples. GPx4 expression levels were*
*measured in HBEC isolated from normal lungs, non-COPD smoker lungs, and*
*mild-severe COPD lungs. Our new experimental result elucidated that GPx4 expression*
*levels were significantly decreased in HBEC isolated from COPD lungs compared to*
*normal lungs. Furthermore, GPx4 expression levels in HBEC were positively correlated*
*with FEV1% in pulmonary function tests. We further confirmed GPx4 reduction in*
*bronchial epithelial cells in severe COPD lungs by means of IHC evaluation. We*
*believe that these additional experimental data supporting the notion that reduced*
*GPx4 levels have a pivotal role in COPD pathogenesis through enhancing lipid*
*peroxidation and subsequent ferroptosis. We hope those experimental results can be*
*sufficient for your major concern. We have added these results in revised Figure 5.*

2. The authors did not address my concern for a mechanistic link between GPX-4
and NCOA4 and even state that "GPx4 and NCOA4 expression levels may not
exist"; and that "Neither NCOA4 nor ferritin was altered by GPx-4
heterozygous mice or the GPX-4 transgenic mice"; This is a fatal flaw for this
paper; how can NCOA4 be proposed as a mechanism for GPX-4 phenotypes in the

murine models when there is no change in NCOA4 expression or ferritin, a downstream
target for NCAO4 in these mice

*We understand your concern regarding the lacking of a mechanistic link between*
*GPX-4 and NCOA4. Although direct association between GPx4 and NCOA4 in terms of*
*expression levels may not apparently exist based on our in vivo experiments, we believe*
*crucial part is concomitant presence of increased NCOA4 and reduced GPX4 in COPD*
*HEBC. Increased NCOA4-induced ferritinophagy is responsible for free iron release,*

*resulting in enhanced hydroxyradical production via Fenton type reaction. Enhanced*
*hydroxyradical production induces excessive lipid peroxidation in the setting of reduced*
*GPX4, leading to exaggerated ferroptosis. Accordingly, we believe the existence of*
*functional link between GPX-4 and NCOA4 with respect to ferroptosis regulation*
*during COPD pathogenesis. Actually, our in vitro experiments demonstrated that*
*NCOA4 knockdown attenuated exaggerated ferroptosis induced by CSE exposure in*
*HBEC with GPX-4 reduction (supplementary Figure10j,k). Taken together, it is likely*
*functional link is more important than association in protein levels, because our*
*experimental results of human samples already showed alteration of both GPX-4 and*
*NCOA4 protein levels, which can be responsible for enhanced ferroptosis during*
*COPD pathogenesis.*

3. The fact that the authors did not know what the background of their mice was
and incorrectly documented this in the original version of this manuscript is as serious
issue. Given the authors use C57/BL6 mice for most of the original experiments and
then use a mixed background for the GPX4 mice is not clean.

*Thank you for your thoughtful suggestion. We apologize our clerical mistake in the*
*revised manuscript. Basically we performed iron accumulation and NCOA4 expression*

*experiment (Fig 3c,d) using control mice of mixed back ground with room air or CS*
*6month exposure in Fig 3e-j experiment. Perls' staining result in Fig 4a was used*
*C57/BL6 mice in the first version manuscript. We reperformed perls' staining with*
*counter staining using control mice (mixed back ground) in Fig 3e-j and also changed*
*Fig4a in the revised manuscript. We corrected method and Figure legend (Fig3).*
*Unfortunately, we couldn't reperform only ICP-MS experiment using mixed background*
*because of technical problem. However, both CS exposed C57BL6 and mixed*
*background mice demonstrated similar experimental results by means of perls' staining.*
*Accordingly, we speculate that this ICP/MS result using C57/BL6 mice is sufficient to*
*support the free iron accumulation as a general response in CS exposed mice lung.*

4. The authors propose to use DFO as a drug to modulate ferroptosis yet they now
state that "DFO did not altered GPx4 and NCOA4 expression"; which
totally diminishes their hypothesis.

*In our understanding, DFO works as a free iron chelator and we believe that the*
*alterations of GPx4 and NCOA4 expression levels may not be necessary in the setting of*
*DFO-mediated ferroptosis attenuation. We have attempted to demonstrate*
*inhibitory role of DFO in CSE-induced lipid peroxidation and ferroptosis*
*via free iron chelation, but not to show DFO-mediated GPx4 and NCOA4 modulation.*
*However, according to this comment, we have analyzed GPx4 and NCOA4 expression*
*levels in response to DFO treatment. NCOA4 expression levels were slightly diminished*
*without significance and GPx4 expression levels were not altered in response to DFO*
*treatment. Although we could not deny the potential regulation of GPx4 and NCOA4*
*expression by free iron levels, DFO treatment could not show significant alteration of*
*those protein levels but showing efficient ferroptosis attenuation.*

Minor

1. If the authors are trying to establish themselves as investigators in the iron field
then please note that the position of the apostrophe in the word; Perl's; the
authors keep putting the apostrophe between the l and the s when in fact it should be
after the s. I am sorry to be pedantic on this point.

*Thank you for your suggestion. We moved the apostrophe after the s.*

2. The authors did not mention total iron as requested.

*We could not analyze total iron concentration because of our technical limitation.*
*However, according to the following your comment "Other readouts for the iron*
*status of the cell would be useful including transferrin receptor activity,*
*Irf1/2, ferroportin etc." , we analyzed transferrin receptor , IRP-1/2, and ferroportin*
*expression level in response to CSE exposure in HBEC (Supplementary Figure 10d).*

Again, thank you for giving us the opportunity to strengthen our manuscript with your
valuable comments and queries. We have worked hard to incorporate your feedback and
hope that these revisions persuade you to accept our submission.

Sincerely,

Shunsuke Minagawa.

Corresponding author/ NCOMMS-18-04233A

Response to REVIEWERS' COMMENTS:

Reviewer #2 (Remarks to the Author):

The authors have sufficiently addressed all concerns from this reviewer.

We thank Reviewer #2 for giving us the opportunity to strengthen our manuscript with your valuable comments and queries.

Reviewer #3 (Remarks to the Author):

The authors have done a good job in answering my concerns.

We wish to express our appreciation to Reviewer #3 for insightful comments, which have helped us significantly improve the paper.

I really like the ferroptosis angle of this paper, however, I am still not convinced about the link between ferroptosis (i.e. GPX-4) and NCOA4 in this model. The siRNA NCOA4 in vivo experiments do not add much and are non-specific and non-selective. I think the NCOA4 mechanism distracts from the overall theme of the paper.

In page17 in the discussion, we included limitation to further reflect reviewer3's opinion ,as below.

Intratracheal NCOA4 siRNA injection in WT mice reduced lipid peroxidation and also reduced CS induced cell death in airway, partially supporting the notion that NCOA4-mediated ferritinophagy has a crucial role in ferroptosis.

“However, this in vivo siRNA experiment has potential limitations with respect to non-specific and non selective method. Further in vivo evidence such as epithelial cell specific NCOA4 knock out mice is needed to support the role of NCOA4 in the development of a COPD phenotype.”

It is intriguing and somewhat concerning that the authors have now generated new data demonstrating that GPX-4 is actually decreased in HBEC cells from COPD patients. This demonstrates the importance of my initial question on focusing in on a particular cell type rather than using whole lung homogenates to make conclusions about a certain pathway/mechanism.

I would suggest that the authors highlight that this is an "epithelial" cell driven phenotype in the title and throughout the text as much as they can.

We changed the title to further reflect reviewer3's opinion that “I really like the ferroptosis angle of this paper “ and “ highlight that this is an "epithelial" cell driven phenotype in the title.”

New title is “ Involvement of cigarette smoke-induced epithelial cell ferroptosis in COPD pathogenesis “

We wish to thank Reviewers again for his or her valuable comments.

Yours sincerely,

Shunsuke Minagawa